# Streaming Kernel PCA with $\tilde{O}(\sqrt{n})$ Random Features

**Enayat Ullah** [†]
enayat@jhu.edu

**Poorya Mianjy** [†]
mianjy@jhu.edu

**Teodor V. Marinov** [†]
tmarino2@jhu.edu

**Raman Arora** [†]
arora@cs.jhu.edu

## Abstract

We study the statistical and computational aspects of kernel principal component analysis using random Fourier features and show that under mild assumptions, $O(\sqrt{n}\log{(n)})$ features suffice to achieve $O(1/\epsilon^2)$ sample complexity. Furthermore, we give a memory efficient streaming algorithm based on classical Oja's algorithm that achieves this rate.

## 1 Introduction

Kernel methods represent an important class of machine learning algorithms that simultaneously enjoy strong theoretical guarantees as well as empirical performance. However, it is notoriously hard to scale them to large datasets due to space and runtime complexity (typically $O(n^2)$ and $O(n^3)$, respectively, for most problems) [Smola and Schölkopf, 1998]. There have been many efforts to overcome these computational challenges, including Nyström method [Williams and Seeger, 2001], incomplete Cholesky factorization [Fine and Scheinberg, 2001], random Fourier features (RFF) [Rahimi and Recht, 2007] and randomized sketching [Yang et al., 2015]. In this paper, we focus on random Fourier features due to its broad applicability to a large class of kernel problems.

In a seminal paper by Rahimi and Recht [2007], the authors appealed to Bochner's theorem to argue that any shift-invariant kernel can be approximated as $k(x, y) \approx \langle z(x), z(y) \rangle$, where the random Fourier feature mapping $z : \mathbb{R}^d \to \mathbb{R}^m$ is obtained by sampling from the inverse Fourier transform of the kernel function. This allows one to invoke fast linear techniques to solve the linear problem in $\mathbb{R}^m$. However, subsequent work analyzing kernel methods based on RFF for learning problems suggests that to achieve the same asymptotic rates (as obtained using the true kernel) on the excess risk, one requires $m = \Omega(n)$ random features [Rahimi and Recht, 2009], which defeats the purpose of using random features from a computational perspective and fails to explain its empirical success.

Last year at NIPS, while Rahimi and Recht won the test-of-time award for their work on RFF [Rahimi and Recht, 2007], Rudi and Rosasco [2017] showed for the first time that at least for the kernel ridge regression problem, under some mild distributional assumptions and for appropriately chosen regularization parameter, one can achieve minimax optimal statistical rates using only $m = O(\sqrt{n}\log{(n)})$ random features. It is then natural to ask if the same holds for other kernel problems.

In this paper, we focus on Kernel Principal Component Analysis (KPCA) [Schölkopf et al., 1998], which is a popular technique for unsupervised nonlinear representation learning. We argue that scalability is an even bigger issue in the unsupervised setting since big data is largely unlabeled. Furthermore, when extending the results from the supervised learning to unsupervised learning we have to deal with additional challenges stemming from the non-convexity of the KPCA problem. We pose KPCA as a stochastic optimization problem and investigate the tradeoff between statistical samples and random features needed to guarantee $\epsilon$-suboptimality on the population objective (aka a small generalization error).

KPCA entails computing the top-$k$ principal components of the data mapped into a Reproducing Kernel Hilbert Space (RKHS) induced by a positive definite kernel [Aronszajn, 1950]. In Schölkopf et al. [1998], authors showed that given a sample of $n$ i.i.d. draws from the underlying distribution, the infinite dimensional problem (over RKHS) can be reduced to a finite dimensional problem (in $\mathbb{R}^n$)

| Algorithm | Reference | Sample complexity | Per-iteration cost | Memory |
|---|---|---|---|---|
| ERM | Shawe-Taylor et al. [2005] | $\tilde{O}(1/\epsilon^2)$ | $\tilde{O}(k/\epsilon^4)$ | $O(1/\epsilon^4)$ |
| | Blanchard et al. [2007][†] | $\tilde{O}(1/\epsilon)$ | $\tilde{O}(k/\epsilon^2)$ | $O(1/\epsilon^2)$ |
| RF-DSG | Xie et al. [2015] | $\tilde{O}(1/\epsilon^2)$ | $\tilde{O}(k/\epsilon^2)$ | $O(k/\epsilon^2)$ |
| RF-ERM | Lopez-Paz et al. [2014] | $\tilde{O}(1/\epsilon^2)$ | $\tilde{O}(k/\epsilon^4)$ | $\tilde{O}(1/\epsilon^4)$ |
| | **Corollary 4.4**[†] | $\tilde{O}(1/\epsilon^2)$ | $\tilde{O}(k/\epsilon^3)$ | $\tilde{O}(1/\epsilon^3)$ |
| RF-Oja | **Corollary 4.4**[†] | $\tilde{O}(1/\epsilon^2)$ | $\tilde{O}(k/\epsilon)$ | $\tilde{O}(k/\epsilon)$ |

Table 1: Comparing different approaches to KPCA in terms of sample complexity, per-iteration computational cost and space complexity. † : Optimistic rates realized under (potentially different) higher-order distributional assumptions (See Corollary 4.3, and Blanchard et al. [2007]).

using the kernel trick. In particular, the solution entails computing the top-$k$ eigenvectors of the kernel matrix computed on the given sample. Statistical consistency of this approach was established in Shawe-Taylor et al. [2005] and further improved in Blanchard et al. [2007]. However, computational aspects of KPCA are less well understood. Note that the eigendecomposition of the kernel matrix alone requires $O(kn^2)$ computation, which can be prohibitive for large datasets. Several recent works have attempted to accelerate KPCA using random features. In Lopez-Paz et al. [2014], authors show that the kernel matrix computed using random features converges to the true kernel matrix in operator norm at a rate of $O(n\sqrt{(\log n)/m})$. In Ghashami et al. [2016], authors extended this guarantee to a streaming setting using the Frequent Direction algorithm [Liberty, 2013] on random features. In a related line of work, Xie et al. [2015] propose a stochastic optimization algorithm based on doubly stochastic gradients with a $1/n$ convergence in the sense of angle between subspaces. However, all these results require $m = \tilde{\Omega}(n)$ random features to guarantee a $O(1/\sqrt{n})$ generalization bound.

More recently, Sriperumbudur and Sterge [2017] studied statistical consistency of ERM with randomized Fourier features. They showed that the top-k eigenspace of the empirical covariance matrix in the random feature space converges to that of the population covariance operator in the RKHS when lifted to the space of square integrable functions, at a rate of $O(1/\sqrt{m} + 1/\sqrt{n})$ [1]. This result suggests that statistical and computational efficiency cannot be achieved at the same time without making further assumptions. In this paper, we assume a spectral decay on the distribution of the data in the feature space to show that we can simultaneously guarantee spectral and computational efficiency for KPCA using random features. Our main contributions are as follows.

1. We study kernel PCA as stochastic optimization problem and show that under mild distributional assumptions, for a wide range of kernels, the empirical risk minimizer (ERM) in the random feature space converges in objective as $O(1/\sqrt{n})$ whenever $m = \Omega(k\sqrt{n}\log{(n)})$, with overall runtime of $O(kn^{\frac{3}{2}}\log{(n)})$.

2. We propose a stochastic approximation algorithm based on classical Oja's updates on random features which enjoys the same statistical guarantees as the ERM above but with better runtime and space requirements.

3. We overcome a key challenge associated with kernel PCA using random features which is to ensure that the output of the algorithm corresponds to a projection operator in the (potentially infinite dimensional) RKHS. We establish that the output of the proposed algorithms converges to a projection operator.

4. In order to better understand the computational benefits of using random features, we also consider the KPCA problem in a streaming setting, where at each iteration, the algorithm is provided with a fresh sample drawn i.i.d. from the underlying distribution and is required to output a solution based on the samples observed so far. In such a setting, comparison with other algorithmic approaches suggests that Oja's algorithm on random Fourier features (see RF-Oja in Table 1) enjoys the best overall runtime as well as superior space complexity.

5. We contribute novel analytical tools that should be useful broadly when designing algorithms for kernel methods based on random features. We provide crucial and novel insights that exploit connections between covariance operators in RKHS and the space of square integrable functions with respect to data distribution. This connection allows us to look

at the kernel *approximation* using random features as an *estimation* problem in the space of square integrable functions, where we appeal to recent results in local Rademacher complexity [Massart, 2000, Bartlett et al., 2002, Blanchard et al., 2007] to yield faster rates.

6. Finally, we provide empirical results on a real dataset to support our theoretical results.

The rest of the paper is organized as follows. In Section 2, we give the problem setup. In Section 3, we provide mathematical preliminaries and introduce the key notation. The main algorithm and the results are in Section 4 and the empirical results are discussed in Section 5.

## 2 Problem setup

Given a random vector $x \in \mathbb{R}^d$ with underlying distribution $\rho$, principal component analysis (PCA) can be formulated as the following stochastic optimization problem [Arora et al., 2012, 2013]:

$$\text{maximize } \mathbb{E}_{x \sim \rho} \langle P, xx^\top \rangle \ \text{ s.t. } P \in \mathcal{P}^k \quad , \tag{1}$$

where $\mathcal{P}^k$ is the set of $d \times d$ rank-$k$ orthogonal projection matrices. Essentially, PCA seeks a $k$-dimensional subspace of $\mathbb{R}^d$ that captures maximal variation with respect to the underlying distribution. It is well understood that the solution to the problem above is given by the projection matrix corresponding to the subspace spanned by the top-$k$ eigenvectors of the covariance matrix $\mathbb{E}\left[xx^\top\right]$.

In most real world applications, however, the data does not have a linear structure. In other words, the underlying distribution may not be well-represented by any low-rank subspace of the ambient space. In such settings, the representations learned using PCA may not be very informative. This motivates the need for non-linear dimensionality reduction methods. For example, in kernel PCA [Schölkopf et al., 1998], a canonical approach for manifold learning, a nonlinear *feature map* lifts the data into a higher (potentially infinite) dimensional Reproducing Kernel Hilbert Space (RKHS), where a low-rank subspace corresponds to a (non-linear) low-dimensional manifold in ambient space. Hence, solving the PCA problem in an RKHS can better capture the complicated nonlinear structure in data.

Formally, given a kernel function $k(\cdot, \cdot) : \mathbb{R}^d \times \mathbb{R}^d \to \mathbb{R}$, KPCA can be formulated as the following stochastic optimization problem:

$$\text{maximize } \mathbb{E}_{x \sim \rho} \langle P, k(x, \cdot) \otimes_{\mathcal{H}} k(x, \cdot) \rangle \ \text{ s.t. } P \in \mathcal{P}^k_{HS(\mathcal{H})} \quad , \tag{2}$$

where $\mathcal{P}^k_{HS(\mathcal{H})}$ is the set of all orthogonal projection operators onto a $k$-dimensional subspace of the RKHS. The solution to the above problem is given by $P^k_C$, the projection operator corresponding to the top-k eigenfunctions of the covariance operator $C := \mathbb{E}_{x \sim \rho}[k(x, \cdot) \otimes_{\mathcal{H}} k(x, \cdot)]$. The primary goal of any KPCA algorithm is then to guarantee generalization, i.e. providing a solution $\widehat{P} \in \mathcal{P}^k_{HS(\mathcal{H})}$ with a small *excess risk*:

$$\mathcal{E}(\widehat{P}) := \mathbb{E}_{x \sim \rho} \langle P^k_C, k(x, \cdot) \otimes_{\mathcal{H}} k(x, \cdot) \rangle - \mathbb{E}_{x \sim \rho} \langle \widehat{P}, k(x, \cdot) \otimes_{\mathcal{H}} k(x, \cdot) \rangle. \tag{3}$$

Given access to i.i.d. samples $\{x_i\}_{i=1}^n \sim \rho$, one approach to solving Problem (2) is **Empirical Risk Minimization (ERM)**, which amounts to finding the top-$k$ eigenfunctions of the empirical covariance operator $\widehat{C} := \frac{1}{n} \sum_{i=1}^n k(x_i, \cdot) \otimes k(x_i, \cdot)$. Using kernel trick, Schölkopf et al. [1998] showed that this problem is equivalent of finding the top-$k$ eigenvectors of the kernel matrix associated with the samples. Alternatively, when approximating the kernel map with random features, Problem (2) reduces to the PCA problem (given in Equation (1)) in the random feature space. Here, we discuss two natural approaches to solve this problem. First, the ERM in the random feature space (called **RF-ERM**), which is given by the top-$k$ eigenvectors of the empirical covariance matrix of data in the feature space. Second, the classical Oja's algorithm (called **RF-Oja**) [Oja, 1982].

Note that while the output of ERM is guaranteed to induce a projection operator in the RKHS of $k(\cdot, \cdot)$, this may not be the case when using RFF (equivalently, when working in the RKHS associated with the approximate kernel map). Therefore, a key technical challenge when designing KPCA algorithm based on RFF is to ensure that the output is close to the set of projection operators in the true RKHS induced by $k(\cdot, \cdot)$, i.e. $d(\widehat{P}, \mathcal{P}^k_{HS(\mathcal{H})})$ is small.

## 3 Mathematical Preliminaries and Notation

In this section, we review basic concepts we need from functional analysis [Reed and Simon, 1972]. We begin with a simple observation that given an underlying distribution on data, and a fixed kernel

map, it induces a distribution on the feature map. We work with this distribution implicitly by considering measurable Hilbert spaces. We denote matrices and Hilbert-Schmidt operators with capital roman letters D, vectors with lower-case roman letters v, and scalars with lower-case letters $a$. We denote operators over the space of Hilbert-Schmidt operators with capital Fraktur letters $\mathfrak{A}$.

**Hilbert space notation and operator norm.** Let $\mathcal{H}$ and $\tilde{\mathcal{H}}$ be two separable Hilbert spaces over fields $\mathbb{F}$ and $\tilde{\mathbb{F}}$ with measures $\mu$ and $\tilde{\mu}$, respectively. Let $\{e_i\}_{i\geq 1}$ and $\{\tilde{e}_i\}_{i\geq 1}$ denote some fixed orthonormal basis for $\mathcal{H}$ and $\tilde{\mathcal{H}}$ respectively. The inner product between two elements $h_1, h_2 \in \mathcal{H}$ is denoted as $\langle h_1, h_2 \rangle_{\mathcal{H}}$, or $\langle h_1, h_2 \rangle_{\mu}$. Similarly, we denote the norm of an element $h \in \mathcal{H}$ as $\|h\|_{\mathcal{H}}$, or $\|h\|_{\mu}$. For $h_1, h_2 \in \mathcal{H}$ the outer product denoted as $h_1 \otimes_{\mathcal{H}} h_2$, or $h_1 \otimes_{\mu} h_2$, is a linear operator on $\mathcal{H}$ that maps any $h_3 \in \mathcal{H}$ to $(h_1 \otimes_{\mathcal{H}} h_2)h_3 = \langle h_2, h_3 \rangle_{\mathcal{H}} h_1$. For a linear operator $D : \mathcal{H} \to \tilde{\mathcal{H}}$, the *operator norm* of D is defined as $\|D\|_2 := \sup\{\|Dh\|_{\tilde{\mathcal{H}}}, h \in \mathcal{H}, \|h\|_{\mathcal{H}} \leq 1\}$.

**Adjoint, Hilbert-Schmidt, and trace-class operators.** The *adjoint* of a linear operator $D : \mathcal{H} \to \tilde{\mathcal{H}}$, is given as the linear operator $D^* : \tilde{\mathcal{H}} \to \mathcal{H}$ such that $\langle Dh, \tilde{h} \rangle_{\tilde{\mathcal{H}}} = \langle h, D^*\tilde{h} \rangle_{\mathcal{H}}$, for all $h \in \mathcal{H}, \tilde{h} \in \tilde{\mathcal{H}}$. A linear operator $D : \mathcal{H} \to \mathcal{H}$ is *self-adjoint* if $D^* = D$. The linear operator $D : \mathcal{H} \to \tilde{\mathcal{H}}$ is *compact* if the image of any bounded set of $\mathcal{H}$ is a relatively compact subset of $\tilde{\mathcal{H}}$. A linear operator $D : \mathcal{H} \to \mathcal{H}$ is a *Hilbert-Schmidt operator* if $\sum_{i\geq 1} \|De_i\|_{\mathcal{H}}^2 = \sum_{i,j\geq 1} \langle De_i, e_j \rangle_{\mathcal{H}}^2 < \infty$. The *Hilbert-Schmidt norm* of D, denoted as $\|D\|_{HS(\mathcal{H})}$ or $\|D\|_{HS(\mu)}$, is defined as $(\sum_{i\geq 1} \|De_i\|_{\mathcal{H}}^2)^{\frac{1}{2}}$. The space of all Hilbert-Schmidt operators on $\mathcal{H}$ is denoted as $HS(\mathcal{H})$. A compact operator $D : \mathcal{H} \to \mathcal{H}$ is *trace-class* if $\|D\|_{\mathcal{L}^1(\mathcal{H})} := \sum_{i\geq 1} \langle (DD^*)^{1/2} e_i, e_i \rangle_{\mathcal{H}} < \infty$, where $\|D\|_{\mathcal{L}^1(\mathcal{H})}$ denotes the nuclear norm of D. For a vector space $\mathcal{X}$, $L^2(\mathcal{X}, \rho)$ denotes the space of square integrable functions with respect to measure $\rho$, i.e $L^2(\mathcal{X}, \rho) = \{f : \mathcal{X} \to \mathbb{R}, \int_{\mathcal{X}} (f(x))^2 d\rho(x) < \infty\}$. $L^2(\mathcal{X}, \rho)$ is a Hilbert space with the inner product denoted as $\langle f, g \rangle_{\rho} := \int_{\mathcal{X}} f(x)g(x)d\rho(x)$, where $f, g \in L^2(\mathcal{X}, \rho)$. The norm induced on $L^2(\mathcal{X}, \rho)$ is denoted as $\|f\|_{\rho} := \langle f, f \rangle_{\rho}^{1/2}$ for $f \in L^2(\mathcal{X}, \rho)$.

**Projection operators, spectral decomposition.** Given a vector space $\mathcal{X}$, let $\mathcal{P}_{\mathcal{X}}^k$ denote the set of rank-$k$ projection operators on $\mathcal{X}$. For a Hilbert-Schmidt operator D over a separable Hilbert space $\mathcal{H}$, let $\lambda_i(D)$ denote its $i^{\text{th}}$ largest eigenvalue. The projection operator associated with the first $k$ eigenfunctions of D is denoted as $P_D^k$; given the spectral decomposition $D = \sum_{i=1}^{\infty} \mu_i \psi_i \otimes \psi_i$, we have that $P_D^k = \sum_{i=1}^{k} \psi_i \otimes \psi_i$. For a finite dimensional vector v, $\|v\|_p$ denotes the $\ell_p$-norm of v. For operators D over finite dimensional spaces, $\|D\|_2$ and $\|D\|_F$ denote the spectral and Frobenius norm of D, respectively. For a metric space $(Y, d)$ and a closed subset $S \subseteq Y$, we denote the distance from $q \in Y$ to S by $d(q, S) = \min_{s \in S} d(q, s)$. In a Hilbert space, $d$ is the underlying metric induced by the respective norm. $[n]$ denotes the set of natural numbers from 1 to $n$.

**Mercer kernels, and random feature maps.** Let $\mathcal{X} \subseteq \mathbb{R}^d$ be a compact (data) domain and $\rho$ be a distribution on $\mathcal{X}$. We are given $n$ independent and identically distributed samples from $\rho$, $\{x_i\}_{i=1}^n \sim \rho^n$. Let $k : \mathcal{X} \times \mathcal{X} \to \mathbb{R}$ be a Mercer kernel with the following integral representation, $k(x, y) = \int_{\Omega} z(x, \omega)z(y, \omega)d\pi(\omega)$. Here, $(\Omega, \pi)$ is the probability space induced by the Mercer kernel. Let $z_{\omega}(\cdot) := z(\cdot, \omega)$. We know that $z_{\omega}(\cdot) \in L^2(\mathcal{X}, \rho)$ almost surely with respect to $\pi$. We draw i.i.d. samples, $\omega_i \sim \pi$, for $i = 1, \ldots, m$, to approximate the kernel function. Let $z(\cdot)$ denote the random feature map, i.e. $z : \mathbb{R}^d \to \mathbb{R}^m, z(x) = \frac{1}{\sqrt{m}} (z_{\omega_1}(x), z_{\omega_2}(x), \ldots, z_{\omega_m}(x))$. Let $\mathcal{F} \subseteq \mathbb{R}^m$ be the linear subspace spanned by the range of z, with the inner product inherited from $\mathbb{R}^m$. The approximate kernel map is denoted as $k_m(\cdot, \cdot)$, where $k_m(x, y) = \langle z(x), z(y) \rangle_{\mathcal{F}}$. Let $\mathcal{H}$ denote the separable RKHS associated with the kernel function $k(\cdot, \cdot)$.

**Assumption 3.1.** The kernel function $k$ is a Mercer kernel (see Theorem A.5) and has the following integral representation, $k(x, y) = \int_{\Omega} z(x, \omega)z(y, \omega)d\pi(\omega) \ \forall x, y \in \mathcal{X}$ where $\mathcal{H}$ is a separable RKHS of real-valued functions on $\mathcal{X}$ with a bounded positive definite kernel $k$. We also assume that there exists $\tau > 1$ such that $|z(x, \omega)| \leq \tau$ for all $x \in \mathcal{X}, \omega \in \Omega$.

Note that $z(x, \cdot)$ are continuous functions because $k(\cdot, \cdot)$ is continuous. Note that when $\mathcal{X}$ is separable and $k(\cdot, \cdot)$ is continuous, $\mathcal{H}$ is separable.

**Definition 3.2.** $C : \mathcal{H} \to \mathcal{H}$ is the covariance operator of the random variables $k(x, \cdot)$ with measure $\rho$, defined as $Cf := \int_{\mathcal{X}} k(x, \cdot)f(x)d\rho(x)$. C is compact and self-adjoint, which implies C has a spectral

decomposition $C = \sum_{i=1}^{\infty} \bar{\lambda}_i \bar{\phi}_i \otimes_{\mathcal{H}} \bar{\phi}_i$, where $\bar{\lambda}_i$'s and $\bar{\phi}_i$'s are the eigenvalues and eigenfunctions of $C$, respectively. The set of eigenfunctions, $\{\bar{\phi}_i\}_{i=1}^{\infty}$, forms a unitary basis for $\mathcal{H}$.

Since we are approximating the kernel $k$ by sampling $m$ i.i.d. copies of $z_\omega$, this implies an approximation to the covariance operator $C$ (in the space $HS(\rho)$) by a sample average of the random linear operators $z_\omega \otimes_\rho z_\omega$. The tools we use to establish concentration require a sufficient spectral decay of the variance of this random operator, which we define next.

**Definition 3.3.** Let $C_1$ denote the random linear operator on $L^2(\mathcal{X}, \rho)$ given by $C_1 = z_\omega \otimes_\rho z_\omega$. Let $C_2 = C_1 \otimes_{HS(\rho)} C_1$ and define the covariance operator of $C_1$ to be $C' = \mathbb{E}_\pi[C_2] - \mathbb{E}_\pi[C_1] \otimes_{HS(\rho)} \mathbb{E}_\pi[C_1]$.

We note that $C'$ can also be interpreted as the fourth moment of the random variable $z_\omega$ in $L^2(\mathcal{X}, \rho)$. The spectrum of $C'$ plays a crucial role in our results through the following key-quantity:

$$\kappa(B_k, k, m) = \inf_{h \geq 0} \left\{ \frac{B_k h}{m} + \sqrt{\frac{k}{m} \sum_{j > h} \lambda_i(C')} \right\}, \quad \text{where } B_k := \frac{\sqrt{\mathbb{E}_\pi\left[\langle z_\omega, z_\omega \rangle_\rho^4\right]}}{\bar{\lambda}_k - \bar{\lambda}_{k+1}} \quad (4)$$

Essentially, we will see that the constant $\kappa(B_k, k, m)$ is the dominating factor when bounding the excess risk, and, therefore, will determine the rate of convergence of our algorithms.

From a practical perspective, working in $HS(\rho)$ is not computationally feasible. However, our approximation to $C$ has a representation in the finite dimensional space $\mathcal{F}$, as defined here.

**Definition 3.4.** $C_m : \mathcal{F} \to \mathcal{F}$ is the covariance operator in $HS(\mathcal{F})$, defined as $C_m := \mathbb{E}_\rho[z(x) \otimes_{\mathcal{F}} z(x)]$. Equivalently, for any $v \in \mathcal{F}$, $C_m v = \int_{\mathcal{X}} \langle z(x), v \rangle z(x) d\rho(x)$. $C_m$ is compact and self-adjoint which implies that $C_m$ has a spectral decomposition $C_m = \sum_{i=1}^{m} \lambda_i \phi_i \otimes_{\mathcal{F}} \phi_i$.

As mentioned at the beginning of the section, our convergence tools work most conveniently when we can incorporate the randomness with respect to $\rho$ in the geometry of the space we study, hence, the need to study $L^2(\mathcal{X}, \rho)$. Since we are essentially dealing with random operators on $\mathcal{F}, \mathcal{H}$ and $L^2(\mathcal{X}, \rho)$, it is most appropriate to also work in the respective spaces of Hilbert-Schmidt operators. Thus, we introduce the *inclusion* and *approximation* operators, which allow us to transition with ease between the aforementioned spaces.

**Definition 3.5.** [Inclusion Operators I and $\mathfrak{I}$] The inclusion operator is defined as
$$I : \mathcal{H} \to L^2(\mathcal{X}, \rho), \ (If) = f, \text{ where } f \in \mathcal{H}.$$

Also, for an operator $D \in HS(\mathcal{H})$ with spectral decomposition $D = \sum_{i=1}^{\infty} \mu_i \psi_i \otimes \psi_i$,

$$\mathfrak{I} : HS(\mathcal{H}) \to HS(\rho), \ \mathfrak{I}D := \sum_{i=1}^{\infty} \mu_i \frac{I\psi_i}{\sqrt{\langle C\psi_i, \psi_i \rangle_{\mathcal{H}}}} \otimes \frac{I\psi_i}{\sqrt{\langle C\psi_i, \psi_i \rangle_{\mathcal{H}}}}.$$

In Lemma A.8 and Lemma A.9 in the appendix, we show that the adjoint of the Inclusion operator I is $I^* : L^2(\mathcal{X}, \rho) \to \mathcal{H}$ given by $(I^*g)(\cdot) = \int k(x, \cdot)g(x)d\rho(x)$, and that $C = I^*I$, $L = II^*$.

**Definition 3.6.** [Approximation Operators A and $\mathfrak{A}$] The Approximation operator A is defined as

$$A : \mathcal{F} \to L^2(X, \rho), (Av)(\cdot) = \langle z(\cdot), v \rangle, \text{ where } v \in \mathcal{F}.$$

For an operator $D \in HS(\mathcal{F})$ with rank $k$ with spectral decomposition $D = \sum_{i=1}^{\infty} \mu_i \psi_i \otimes \psi_i$, let $\Psi$ be the matrix with eigenvectors $\psi_i$ as columns and let $\Phi$ be the matrix with eigenvectors of $C_m$ as columns (see Definition 3.4). Define
$$R^* = \operatorname*{arg\,min}_{R^\top R = RR^\top = I} \|\Psi R - \Phi\|_{\mathcal{F}}^2, \ \tilde{\Psi} := \Psi R^*.$$

Let $\tilde{\psi}_i$ be the $i^{th}$ column of $\tilde{\Psi}$, define

$$\mathfrak{A} : HS(\mathcal{F}) \to HS(\rho), \ \mathfrak{A}D := \sum_{i=1}^{k} \mu_i \frac{A\tilde{\psi}_i}{\sqrt{\langle C_m \tilde{\psi}_i, \tilde{\psi}_i \rangle_{\mathcal{F}}}} \otimes_\rho \frac{A\tilde{\psi}_i}{\sqrt{\langle C_m \tilde{\psi}_i, \tilde{\psi}_i \rangle_{\mathcal{F}}}}.$$

In Lemma A.11 and Lemma A.12, we show that the adjoint of the Approximation Operator is $A^* : L^2(X, \rho) \to \mathcal{F}, (A^*f)_i = \int_{\mathcal{X}} f(x) z_{\omega_i}(x) d\rho(x)$, and $C_m = A^*A, L_m = AA^*$.

We note that the definition of the approximation operator $\mathfrak{A}$ requires knowledge of the covariance matrix $C_m$ to find the optimal rotation matrix $R^*$, but this is solely for the purpose of analysis and is not used in the algorithm in any form.

The following definition enables us to bound the excess risk in $HS(\mathcal{H})$ (Section B in the appendix).

**Definition 3.7.** [Operator $\mathfrak{L}$] Let $\tilde{P} \in HS(\mathcal{F})$. Let $\mathfrak{A}\tilde{P} = \sum_{i=1}^k \tilde{p}_i \otimes_\rho \tilde{p}_i$ be $\tilde{P}$ lifted to $HS(\rho)$. Consider the equivalence relation $p_i \sim p_j$ if $L^{1/2}p_i = L^{1/2}p_j$. Let $[p_i]$ be the equivalence class such that $L^{1/2}p_i = \tilde{p}_i$. The operator $\mathfrak{L} : HS(\mathcal{F}) \to HS(\mathcal{H})$ is defined as $\mathfrak{L}\widehat{P} = \sum_{i=1}^k I^*p_i \otimes_{\mathcal{H}} I^*p_i$. Here $I^*$ is the restriction of the operator $I^*$ to the quotient space $L^2(\mathcal{X}, \rho)/ \sim$.

The quotient space in the definition above is with respect to the kernel of L, i.e., $L^2(\mathcal{X}, \rho)/ \sim \equiv L^2(\mathcal{X}, \rho)/\ker(L)$. This quotient is benign since the optimal solution to our optimization problem lives in the range of L and intuitively we can disregard any components in the kernel of L.

Finally, to conclude the section we give a visual schematic in Figure 1 to help the reader connect different spaces. To summarize, the key spaces of interest are the data domain $\mathcal{X}$, the RKHS $\mathcal{H}$ of the kernel map $k(\cdot, \cdot)$, and the feature space $\mathcal{F}$ obtained via random feature approximation. The space $L^2(\mathcal{X}, \rho)$ consists of functions over the data domain $\mathcal{X}$ that are square integrable with respect to the data distribution $\rho$. The space $L^2(\mathcal{X}, \rho)$ allows us to embed objects from different spaces into a common space so as to compare them. Specifically, we map functions from $\mathcal{H}$ to $L^2(\mathcal{X}, \rho)$ via the inclusion operator I, and vectors from $\mathcal{F}$ to $L^2(\mathcal{X}, \rho)$ via the approximation operator A. $I^*$ and $A^*$ denote the adjoints of I and A, respectively. The space of Hilbert-Schmidt operators on $\mathcal{H}, \mathcal{F}$ and $L^2(\mathcal{X}, \rho)$, are denoted by $HS(\mathcal{H}), HS(\mathcal{F})$ and $HS(\rho)$, respectively. Analogous to I and A, $\mathfrak{I}$ maps operators from $HS(\mathcal{H})$ to $HS(\rho)$, and $\mathfrak{A}$ maps operators from $HS(\mathcal{F})$ to $HS(\rho)$, respectively. Specifically, these are essentially constructed by mapping eigenvectors of operators

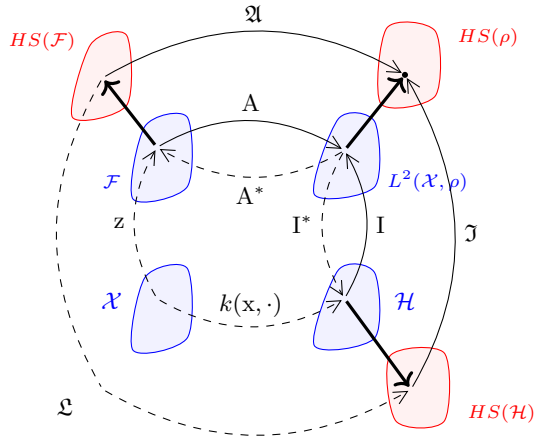

Figure 1: Maps between the data domain ($\mathcal{X}$), space of square integrable functions on $\mathcal{X}$ ($L^2(\mathcal{X}, \rho)$), the RKHS of kernel $k(\cdot, \cdot)$, and RKHS of the approximate feature map, as well as maps between Hilbert-Schmidt operators on these spaces.

via I and A respectively. The above mappings thus allow us to embed operators in the common space, i.e., $HS(\rho)$ and to bound estimation and approximation errors. However, the problem of Kernel PCA is formulated in $HS(\mathcal{H})$ and bounds in $HS(\rho)$ are therefore not sufficient. To this end, we establish an equivalence between kernel PCA in $HS(\mathcal{H})$ and $HS(\rho)$. We use the map $\mathfrak{A}$ and the established equivalence to get $\mathfrak{L}$, which maps operators from $HS(\mathcal{F})$ to $HS(\mathcal{H})$. We encourage the reader to go through Sections A and B in the appendix for a gentler and a more rigorous presentation.

## 4 Main Results

Recall that our primary goal is to study the generalization behaviour of algorithms solving KPCA using random features. Rather than stick to a particular algorithm, we define a class of algorithms that are suitable to the problem. We characterize this class as follows.

**Definition 4.1** (Efficient Subspace Learner (ESL)). Let $\mathcal{A}$ be an algorithm which takes as input $n$ points from $\mathcal{F}$ and outputs a rank-k projection matrix over $\mathcal{F}$. Let $\widehat{P}_{\mathcal{A}}$ denote the output of the algorithm $\mathcal{A}$ and $\widehat{P}_{\mathcal{A}} = \tilde{\Phi}\tilde{\Phi}^\top$ be an eigendecompostion of $\widehat{P}_{\mathcal{A}}$. Let $\Phi_k^\perp$ be an orthogonal matrix corresponding to the orthogonal complement of the top $k$ eigenvectors of $C_m$. We say that algorithm $\mathcal{A}$ is an Efficient Subspace Learner if the following holds with probability at least $1 - \delta$,

$$\left\| (\Phi_k^\perp)^\top \tilde{\Phi} \right\|_F^2 \leq \frac{q_{\mathcal{A}}^{\rho;\pi}(1/\delta, \log(m), \log(n))}{n},$$

---
**Algorithm 1** KPCA with Random Features (Meta Algorithm)
---
**Input:** Training data $X = \{x_i\}_{i=1}^n$

**Output:** $\widehat{P}_{\mathcal{A}}$

  1: Obtain Training data $X = \{x_t\}_{i=1}^n$ in a batch or stream
  2: Sample $\omega_i \sim \pi$, i.i.d, $i = 1$ to $m$
  3: $Z \leftarrow \texttt{RandomFeatures}(X, \{\omega_i\}_{i=1}^m)$
  4: $\widehat{P}_{\mathcal{A}} \leftarrow \mathcal{A}(Z)$                    //$\mathcal{A}$ is an Efficient Subspace Learner, Definition 4.1
---

where $q_{\mathcal{A}}^{\rho,\pi}$ is a function given the triple $(\mathcal{A}, \rho, \pi)$ which has polynomial dependence on $1/\delta$, $\log(m)$ and $\log(n)$. For notational convenience, we drop superscripts from $q_{\mathcal{A}}^{\rho,\pi}$ and write it as $q_{\mathcal{A}}$ henceforth.

Intuitively, an ESL is an algorithm which returns a projection onto a $k$-dimensional subspace such that the angle between the subspace and the space spanned by the top $k$ eigenvectors of $C_m$ decays at a sub-linear rate with the number of samples. Our guarantees are in terms of any algorithm which belongs to this class. Algorithm 1 gives a high-level view of the algorithmic routine. To discuss the associated computational aspects, we instantiate this with two specific algorithms, ERM and Oja's algorithm, and show how the result looks in terms of their algorithmic parameters. Similar results can be obtained for other ESL algorithms such as $\ell_2$-RMSG [Mianjy and Arora, 2018]. We now give the main theorem of the paper which characterizes the excess risk of an ESL.

**Theorem 4.2** (Main Theorem). *Let $\mathcal{A}$ be an efficient subspace learner. Let $\widehat{P}_{\mathcal{A}}$ be the output of $\mathcal{A}$ run with $m$ random features on $n \geq \frac{2\lambda_1^2 q_{\mathcal{A}}(2/\delta,\log(m),\log(n))^2}{\lambda_k^2(\sqrt{2}-1)}$ points, where $\lambda_i$ is the $i^{th}$ eigenvalue of $C_m$. Then, with probability at least $1-\delta$ it holds that*

*(a). $\mathcal{E}(\mathfrak{L}\widehat{P}_{\mathcal{A}}) \leq 24\kappa(B_k,k,m) + \frac{\log(\delta/2)+7B_k}{m} + \sqrt{\frac{q_{\mathcal{A}}(2/\delta,\log(m),\log(n))}{n}}$,*

*(b). $d\left(\mathfrak{L}\widehat{P}_{\mathcal{A}}, \mathcal{P}_{HS(\mathcal{H})}^k\right) \leq \sqrt{\frac{q_{\mathcal{A}}(2/\delta,\log(m),\log(n))}{n}}$.*

A few remarks are in order. First, as we forewarned the reader in Section 3, the error bound is dominated by the additive term $\kappa(B_k, k, m)$. This, in a sense, determines the hardness of the problem. As we will see, under appropriate assumptions on data distribution in the feature space, this term can be bounded by something that is in $O(1/m)$. Second, the output of our algorithm, $\mathfrak{L}\widehat{P}$, need not be a projection operator in the RKHS. This is precisely why we need to bound the difference between $\mathfrak{L}\widehat{P}$ and the set of all projection operators in $HS(\mathcal{H})$, which we see is of the order $O(1/\sqrt{n})$. Third, note that the dependence on the number of random features is at worst poly-logarithmic. From part $(b)$ of Theorem 4.2, it is easy to see that if we project $\mathfrak{L}\widehat{P}_{\mathcal{A}}$ to the set of rank $k$ projection operators in $HS(\mathcal{H})$, we get the same rate of convergence. This is presented as Corollary C.12 in the appendix.

Next, we characterize "easy" instances of KPCA problems under which we are guaranteed a fast rate. Specifically, we show that if the decay of the spectrum of the fourth order moment, $C'$, of $z_\omega$, is exponential, then the dominating factor, $\kappa(B_k, k, m)$ is in $O(1/m)$. Then, optimizing the number of random features w.r.t. the sample complexity term gives us the following result.

**Corollary 4.3** (Main - Good decay). *Along with the assumptions and notation of Theorem 4.2, if the spectrum of the operator $C'$ has an exponential decay, i.e., $\lambda_j(C') = \alpha^j$ for some $\alpha < 1$, then with $m = O(\sqrt{n}\log(n))$ random features, we have*

$$\mathcal{E}(\mathfrak{L}\widehat{P}_{\mathcal{A}}) \leq \frac{cB_k}{\sqrt{n}} + \frac{c'(k + \log(\delta/2) + 7B_k)}{\sqrt{n}\log(n)} + \sqrt{\frac{q_{\mathcal{A}}(2/\delta, \log(m), \log(n))}{n}},$$

*where $c$ and $c'$ are universal constants.*

Finally, we instantiate the above corollary with two algorithms, namely ERM and Oja's algorithm.

**Corollary 4.4** (ERM and Oja). *With the same assumptions and notation as in Corollary 4.3,*

*(a). **RF-ERM** is an ESL with $q_{ERM}(1/\delta, \log(m), \log(n)) = \frac{k\lambda_1\tau^2}{\text{gap}^2}\log\left(\frac{\delta}{2m}\right)^2$.*

*(b). **RF-Oja** is an ESL with $q_{oja}(1/\delta, \log(m), \log(n)) = \tilde{\Theta}\left(\frac{\Lambda}{\text{gap}^2}\right)$, where $\Lambda = \sum_{i=1}^k \lambda_i$.*

*where $\text{gap} := \lambda_k(C_m) - \lambda_{k+1}(C_m)$.*

**Error Decomposition:** There are two sources of error when solving KPCA using random features – the estimation error ($\epsilon_e$) resulting from the fact that we have access to the distribution only through an i.i.d. sample, and approximation error ($\epsilon_a$) resulting from the approximate feature map. Therefore, to get a better handle on the excess error, we decompose it as follows.

$$\mathcal{E}(\mathfrak{L}\widehat{\mathrm{P}}_{\mathcal{A}}) = \underbrace{\langle \mathrm{P}_{\mathrm{C}}^k, \mathrm{C} \rangle_{HS(\mathcal{H})} - \langle \mathfrak{L}\mathrm{P}_{\mathrm{C}_m}^k, \mathrm{C} \rangle_{HS(\mathcal{H})}}_{\epsilon_a : \text{Approximation Error}} + \underbrace{\langle \mathfrak{L}\mathrm{P}_{\mathrm{C}_m}^k, \mathrm{C} \rangle_{HS(\mathcal{H})} - \langle \mathfrak{L}\widehat{\mathrm{P}}_{\mathcal{A}}, \mathrm{C} \rangle_{HS(\mathcal{H})}}_{\epsilon_e : \text{Estimation Error}}.$$

The main idea behind controlling the approximation error is to interpret it as the error incurred in eigenspace estimation in $L^2(\mathcal{X}, \rho)$, and then use local Rademacher complexity to get faster rates. In the context of Kernel PCA, this technique was first used by Blanchard et al. [2007] which allowed them to get sharper $O(1/n)$ excess risk. The estimation error is controlled by the definition of our lifting map $\mathfrak{A}$ together with the convergence rate implicit in the definition of an ESL. Below, we guide the reader through the main steps taken to bound each of the error terms.

**Bounding the Approximation error:** Using simple algebraic manipulations, we can show that the approximation error is exactly the error incurred by the ERM in estimating the top $k$ eigenfunctions of the kernel integral operator L using $m$ samples drawn from $\pi$. This problem of eigenspace estimation is well studied in the literature and has optimal statistical rates of $O\left(1/\sqrt{m}\right)$ [Zwald and Blanchard, 2006]. This appears to be a key bottleneck and reinforces the view that the use of random features cannot provide computational benefits – it suggests $m = \Omega(n)$ random features are required to get a $O\left(1/\sqrt{n}\right)$ rate. However, these rates are conservative when viewed in the sense of excess risk. This has been extensively studied in empirical process theory and one of the primary techniques to get sharper rates is the use of local Rademacher complexity [Bartlett et al., 2002]. The key idea is to show that around the best hypothesis in the class, variance of the empirical process is bounded by a constant times the mean of the difference from the best hypothesis (see Theorem F.3). This technique was used in the context of Kernel PCA by Blanchard et al. [2007] to get fast $O(1/m)$ rates. We now state Lemma 4.5 which bounds the approximation error, the proof of which is deferred to appendix.

**Lemma 4.5** (Approximation Error). *With probability at least $1 - \delta$, we have*

$$\epsilon_a \leq 24\kappa(B_k, k, m) + \frac{11\tau^2 \log(\delta) + 7B_k}{m}.$$

**Bounding the Estimation error:** Since the objective with respect to the inner product in $HS(\rho)$ equals the objective with respect to the inner product in $HS(\mathcal{H})$ (See Lemma B.4), we focus on bounding the estimation error in $L^2(\mathcal{X}, \rho)$. Using a Cauchy-Schwartz type of inequality in $HS(\rho)$, we see that it is enough to bound the difference $\|\mathfrak{A}\mathrm{P}_{\mathrm{C}_m}^k - \mathfrak{A}\widehat{\mathrm{P}}_{\mathcal{A}}\|_{HS(\rho)}$. We can do this in two steps – bound the error $\|\mathrm{P}_{\mathrm{C}_m}^k - \widehat{\mathrm{P}}_{\mathcal{A}}\|_F$ (we already have this from the ESL guarantee) and construct $\mathfrak{A} : HS(\mathcal{F}) \rightarrow HS(\rho)$. We already have a lifting from $\mathcal{F}$ to $L^2(\mathcal{X}, \rho)$ in the form of A. The natural attempt to lift an operator on $\mathcal{F}$ would be by lifting and appropriately rescaling its eigenfunctions. Since the eigendecomposition of $\widehat{\mathrm{P}}_{\mathcal{A}}$ is not unique, we need to choose an appropriate one to be lifted. Since the goal of $\mathfrak{A}$ is to preserve distances between operators, we choose the unique eigendecomposition for which the distance $\sum_{i=1}^{k} \|\mathrm{U}_i - \phi_i\|_2^2$ is minimized. Notice that the lifting operator $\mathfrak{A}$ depends on the eigendecomposition of $\mathrm{C}_m$, which can not be obtained in practice. This is not a problem, because $\mathfrak{A}$ is only used for the purposes of showing the main result and is not part of the proposed algorithms. We now state Lemma 4.6 which bounds the estimation error.

**Lemma 4.6** (Estimation Error). *With the same assumptions as Theorem 4.2, the following holds with probability at least $1 - \delta$,*

$$\epsilon_e \leq \frac{\lambda_1^2}{(\sqrt{2} - 1)} \sqrt{\sum_{i=1}^{k} \left(\frac{2\lambda_i + 4\lambda_1}{\lambda_i^2}\right)^2 \frac{q_{\mathcal{A}}(1/\delta, \log(m), \log(n))^2}{n}}.$$

## 5  Experiments

The goal of this section is to provide empirical evidence supporting our theoretical findings in Section 4. As we motivated in Section 2, the success of an algorithm is measured in terms of it's *generalization* ability, i.e. the variance captured by the output of the algorithm on the unseen data[2].

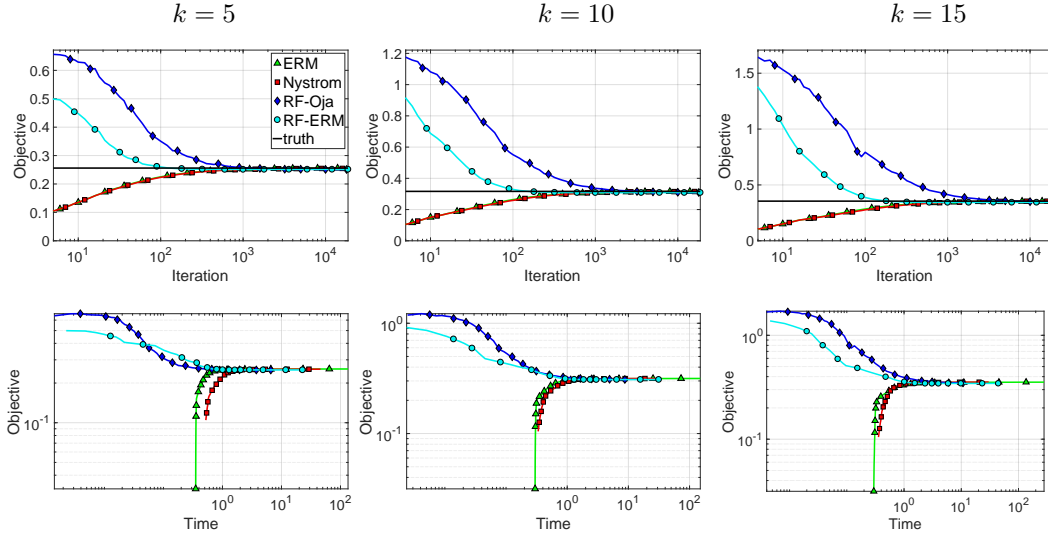

Figure 2: Comparisons of ERM, Nyström, Oja+RFF, and Oja+ERM for KPCA on the MNIST dataset, in terms of the objective value as a function of iterations (top) and as a function of CPU runtime (bottom).

We perform experiments on the MNIST dataset that consists of 70K samples, partitioned into a training, tuning, and a test set of sizes 20K, 10K, and 40K, respectively. We use a fixed kernel in all our experiments, since we are not concerned about model selection here. In particular, we choose the RBF kernel $k(\mathrm{x}, \mathrm{x}') = \exp\left(-\|\mathrm{x} - \mathrm{x}'\|^2 / 2\sigma^2\right)$ with bandwidth parameter $\sigma^2 = 50$. The bandwidth is chosen such that ERM converges in objective within observing few thousands training samples. The objective of the ERM[3] is used as the baseline. Furthermore, to evaluate the computational speedup gained by using random features, we compare against Nyström method [Drineas and Mahoney, 2005] as a secondary baseline. In particular, upon receiving a new sample, we do a full Nyström approximation and ERM on the set of samples observed so far. Finally, empirical risk minimization (RF-ERM) and Oja's algorithm (RF-Oja) are used with random features to verify the theoretical results presented in Corollary 4.4.

Figure 2 shows the population objective as a function of iteration (top row) as well as the total runtime[4] (bottom row). Each curve represents an average over 100 runs of the corresponding algorithm on training samples drawn independently and uniformly at random from the whole dataset. Number of random features and the size of Nyström approximation are set to 750 and 100, respectively. We note:

- As predicted by Corollary 4.4, for both RF-ERM and RF-Oja, $\sqrt{n} \log(n) \approx 750$ random features is sufficient to achieve the same suboptimality as that of ERM.

- The performance of ERM is similar to that of RF-Oja and RF-ERM in terms to overall runtime. However, due to larger space complexity of $O(n^2)$, ERM becomes infeasible for large-scale problems; this makes a case for streaming/stochastic approximation algorithms.

Finally, we note that the iterates of RF-ERM and RF-Oja *reduce* the objective as they approach from above to the *maximizer* of the population objective. Although it might seem counter-intuitive, we note that the output of RF-ERM and RF-Oja are not necessarily projection operators. Hence, they can achieve higher objective than the maximum. However, as guaranteed by Corollary 4.4, the output of both algorithms will converge to a projection operator as more training samples are introduced.

## Acknowledgements

This research was supported in part by NSF BIGDATA grant IIS-1546482.

## Footnotes

† Department of Computer Science, Johns Hopkins University, Baltimore, MD 21204

[1]While our paper was under review, Sriperumbudur and Sterge [2017], which initially focused on statistical consistency of kernel PCA with random features, was replaced by Sriperumbudur and Sterge [2018], with a new title and focus on computational and statistical tradeoffs of KPCA much like our paper.

[2]Details on how we evaluate objective for RF-ERM/Oja are deferred to Section E due to space limitations.

[3]The kernel matrix is computed in an online fashion for computational efficiency

[4]Runtime is recorded in a controlled environment; each run executed on identical unloaded compute node.

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
