[Supplementary Material · supplementary_3643.pdf]

The Appendix is divided into the following six sections,

# A  Preliminaries and Structural Results

We begin this section by giving some definitions and structural theorems that are required for the proofs. We first introduce a few additional notation. Consider two separable Hilbert spaces $\mathcal{H}_1$ and $\mathcal{H}_2$. For an operator $\mathcal{D} : \mathcal{H}_1 \to \mathcal{H}_2$, we use $\|\mathcal{D}\|_{\mathcal{L}^p(\mathcal{H}_1, \mathcal{H}_2)}$ to denote its $p^{\text{th}}$ schatten norm, assuming that it is finite. We omit the $p$ when we talk about about Hilbert-Schmidt norm i.e. $p = 2$.

## A.1  Covariance operators $\mathrm{C}$ and $\mathrm{C}_m$

The covariance operators in the RKHS and the random feature space are defined as follows:

**Definition A.1.** $\mathrm{C} : \mathcal{H} \to \mathcal{H}$ is the co-variance operator of the random variables $k(\mathrm{x}, \cdot)$ with measure $\rho$, defined as:

$$\mathrm{C}f := \int_{\mathcal{X}} k(\mathrm{x}, \cdot) f(\mathrm{x}, t) d\rho(\mathrm{x}, t)$$

$\mathrm{C}$ is compact and self-adjoint, which implies $\mathrm{C}$ has a spectral decomposition as follows:

$$\mathrm{C} = \sum_{i=1}^{\infty} \bar{\lambda}_i \bar{\phi}_i \otimes_{\mathcal{H}} \bar{\phi}_i$$

where $\bar{\lambda}_i, \bar{\phi}_i$'s are the eigenvalues and eigenfunctions of $\mathrm{C}$. Also, $\bar{\phi}_i$ are a unitary basis for $\mathcal{H}$.

**Definition A.2.** $\mathrm{C}_m : \mathcal{F} \to \mathcal{F}$ is the covariance operator in the random feature space, defined as

$$\mathrm{C}_m := \mathbb{E}_{\rho} \left[ \mathrm{z}(\mathrm{x}, t) \otimes_{\mathcal{F}} \mathrm{z}(\mathrm{x}, t) \right]$$

Equivalently, for any $\beta \in \mathcal{F}$, $\mathrm{C}_m \beta = \int_{\mathcal{X}} \langle \mathrm{z}(\mathrm{x}, t), \beta \rangle \, \mathrm{z}(\mathrm{x}, t) d\rho(\mathrm{x}, t)$.

$\mathrm{C}_m$ is compact and self-adjoint which implies that $\mathrm{C}_m$ has a spectral decomposition as follows:

$$\mathrm{C}_m = \sum_{i=1}^{m} \lambda_i \phi_i \otimes_{\mathcal{F}} \phi_i$$

The kernel integral operators and its approximation based on random features are defined as follows:

**Definition A.3.** The kernel integral operator $\mathrm{L} : L^2(\mathcal{X}, \rho) \to L^2(\mathcal{X}, \rho)$ is defined as follows:

$$\mathrm{L}g = \int_{\mathcal{X}} k(\mathrm{x}, \cdot) g(\mathrm{x}) d\rho(\mathrm{x}) \ \forall \ g \in L^2(\mathcal{X}, \rho)$$

**Definition A.4.** $\mathrm{L}_m : L^2(\mathcal{X}, \rho) \to L^2(\mathcal{X}, \rho)$ is the (approximated) kernel integral operator, defined as:

$$(\mathrm{L}_m g)(\cdot) = \int_{\mathcal{X}} k_m(\mathrm{x}, \cdot) g(\mathrm{x}) d\rho(\mathrm{x})$$

We state the classical Mercer's and Bochner's theorems for completeness.

**Theorem A.5** (Mercer's Theorem). *For every positive definition kernel $k(\cdot, \cdot) : \mathcal{X} \times \mathcal{X} \to \mathbb{R}$, there exits a set $\Omega$ with measure $\pi$, and functions $\varpi(\cdot) : \mathcal{X} \times \Omega \to \mathbb{R}$ such that the kernel has an integral representation of the following form,*

$$k(\mathrm{x}, \mathrm{y}) = \int_{\Omega} z(\mathrm{x}, \omega) z(\mathrm{y}, \omega) d\pi(\omega) \ \forall \ \mathrm{x}, \mathrm{y} \in \mathcal{X}$$

In particular, for shift-invariant kernels, we have

**Theorem A.6** (Bochner's Theorem Rudin [2017])**.** *A continuous, real-valued, symmetric shift-invariant kernel $k : \mathcal{X} \times \mathcal{X} \to \mathbb{R}$ is a positive-definite kernel if and only if there exits a non-negative measure $\pi(\omega)$ such that $k(x-y) = \int_{\mathcal{X}} e^{i\omega^{\top}(x-y)} d\pi(\omega)$ i.e the inverse Fourier transform of $k(x-y)$*

For a comprehensive list of kernels with their Fourier transform see Table 1 of Xie et al. [2015].

We now define operators to lift functions and operators from and into different spaces. These are crucially used in the analysis of the algorithm. See Figure 3 in for a schematic of the lifting operators.

## A.2 Inclusions operators $I, \mathfrak{I}$

We first recall the definitions of Inclusion operators $I, \mathfrak{I}$.

**Definition A.7.** [Inclusion Operators I and $\mathfrak{I}$] The inclusion operator is defined

$$I : \mathcal{H} \to L^2(\mathcal{X}, \rho), \ (If) = f, \ \text{where } f \in \mathcal{H}$$

Also, for an operator $D \in HS(\mathcal{H})$ with spectral decomposition $D = \sum_{i \in I \subset \mathbb{R}} \mu_i \psi_i \otimes \psi_i$,

$$\mathfrak{I} : HS(\mathcal{H}) \to HS(\rho), \ \mathfrak{I}D := \sum_{i \in I \subset \mathbb{R}} \mu_i \frac{I\psi_i}{\sqrt{\langle C\psi_i, \psi_i \rangle_{\mathcal{H}}}} \otimes \frac{I\psi_i}{\sqrt{\langle C\psi_i, \psi_i \rangle_{\mathcal{H}}}}$$

In Proposition A.8, we show that the adjoint of the Inclusion operator I is

$$I^* : L^2(\mathcal{X}, \rho) \to \mathcal{H}, (I^*g)(\cdot) = \int k(x, \cdot)g(x)d\rho(x).$$

Moreover, In Proposition A.9 we show that the covariance operator and the kernel integral operator can be expressed in terms of I and I∗ as $C = I^*I$ and $L = II^*$.

**Proposition A.8.** *The following holds with regard to the inclusion operator,*

*(a). The adjoint of the Inclusion operator $I$ is given by $(I^*g)(\cdot) = \int_{\mathcal{X}} k(x, \cdot)g(x)d\rho(x)$.*

*(b). $I$ and $I^*$ are Hilbert-Schmidt.*

*Proof of Proposition A.8. (a).* We first show that the adjoint of the Inclusion operator I is given by $(I^*g)(\cdot) = \int_{\mathcal{X}} k(x, \cdot)g(x)d\rho(x)$. For $f \in \mathcal{H}$ and $g \in L^2(\mathcal{X}, \rho)$, we have that

$$
\begin{aligned}
\langle If, g \rangle_{\rho} &= \langle f, g \rangle_{\rho} & \text{(Definition of I)} \\
&= \int_{\mathcal{X}} f(x)g(x)d\rho(x) \\
&= \int_{\mathcal{X}} \langle k(x, \cdot), f \rangle_{\mathcal{H}} g(x)d\rho(x) & \text{(Reproducing property)} \\
&= \int_{\mathcal{X}} \langle k(x, \cdot)g(x), f \rangle_{\mathcal{H}} d\rho(x) & \text{(Linearity of inner product)} \\
&= \left\langle \int_{\mathcal{X}} k(x, \cdot)g(x)d\rho(x), f \right\rangle_{\mathcal{H}} & \text{(Fubini's Theorem)} \\
&= \langle I^*g, f \rangle_{\mathcal{H}}
\end{aligned}
$$

*(b).* Let $\{\bar{e}_i\}_{i=1}^{\infty}$ be an orthonormal basis for $\mathcal{H}$. We have,

$$\|\mathrm{I}\|_{\mathcal{L}(\mathcal{H},\rho)}^2 = \sum_{i=1}^{\infty} \|\mathrm{I}\bar{e}_i\|_{\rho}^2 \qquad \text{(Pythagoras Theorem)}$$

$$= \sum_{i=1}^{\infty} \|\bar{e}_i\|_{\rho}^2 \qquad \text{(Definition 3.5)}$$

$$= \sum_{i=1}^{\infty} \int_{\mathcal{X}} \langle k(\mathrm{x},\cdot), \bar{e}_i \rangle_{\mathcal{H}}^2 \, d\rho(\mathrm{x}) \qquad \text{(Reproducing Property)}$$

$$= \int_{\mathcal{X}} \sum_{i=1}^{\infty} \langle k(\mathrm{x},\cdot), \bar{e}_i \rangle_{\mathcal{H}}^2 \, d\rho(\mathrm{x}) \qquad \text{(Fubini's Theorem)}$$

$$= \int_{\mathcal{X}} k(x,x) d\rho(\mathrm{x}) \leq \tau^2 < \infty \qquad \text{(Assumption 3.1)}$$

For the adjoint $\mathrm{I}^*$, we have $\|\mathrm{I}^*\|_{\mathcal{L}(\rho,\mathcal{H})} = \|\mathrm{I}\|_{\mathcal{L}(\mathcal{H},\rho)} < \infty$ $\qquad\qquad\square$

**Proposition A.9.** *The following properties hold,*

   *(a). The covariance operator and the kernel integral operator satisfy* $\mathrm{C} = \mathrm{I}^*\mathrm{I}$ *and* $\mathrm{L} = \mathrm{II}^*$ *respectively.*

   *(b).* $\mathrm{C}$ *and* $\mathrm{L}$ *are trace-class*

*Proof of Proposition A.9. (a).* We first show that $\mathrm{C} = \mathrm{I}^*\mathrm{I}$. For any $f \in L^2(\mathcal{X},\rho)$, we have

$$\mathrm{I}^*\mathrm{I}f = \mathrm{I}^* f \qquad \text{(Definition A.7)}$$

$$= \int_{\mathcal{X}} k(\mathrm{x},\cdot) f(\mathrm{x}) d\rho(\mathrm{x}) \qquad \text{(Proposition A.8)}$$

$$= \mathrm{C}f \qquad \text{(Definition A.1)}$$

We now show that $\mathrm{L} = \mathrm{II}^*$. For any $g \in L^2(\mathcal{X},\rho)$, we have

$$\mathrm{II}^*g = \mathrm{I}\left( \int_{\mathcal{X}} k(\mathrm{x},\cdot) g(\mathrm{x}) d\rho(\mathrm{x}) \right) \qquad \text{(Proposition A.8)}$$

$$= \int_{\mathcal{X}} k(\mathrm{x},\cdot) g(\mathrm{x}) d\rho(\mathrm{x}) \qquad \text{(Definition A.7)}$$

$$= \mathrm{L}g$$

*(b).* Now we show that $\mathrm{C}$ and $\mathrm{L}$ are trace-class.

$$\|\mathrm{C}\|_{\mathcal{L}^1(\mathcal{H})} = \|\mathrm{I}^*\mathrm{I}\|_{\mathcal{L}^1(\mathcal{H})} \qquad \text{(Proposition A.9)}$$

$$= \|\mathrm{I}\|_{\mathcal{L}^2(\mathcal{H})}^2 < \infty \qquad \text{(Proposition A.8)}$$

Similarly,

$$\|\mathrm{L}\|_{\mathcal{L}^1(\rho)} = \|\mathrm{II}^*\|_{\mathcal{L}^1(\rho)} \qquad \text{(Proposition A.9)}$$

$$= \|\mathrm{I}\|_{\mathcal{L}^2(\mathcal{H})}^2 < \infty \qquad \text{(Proposition A.8)}$$

$$\square$$

## A.3 Approximation operators $\mathrm{A}$ and $\mathfrak{A}$

We first recall the definitions of approximation operators $\mathrm{A}$ and $\mathfrak{A}$.

**Definition A.10.** [Approximation Operators $\mathrm{A}$ and $\mathfrak{A}$] The Approximation operator $\mathrm{A}$ is defined as

$$\mathrm{A} : \mathcal{F} \to L^2(\mathrm{X},\rho), (\mathrm{A}\mathrm{v})(\cdot) = \langle z(\cdot), \mathrm{v} \rangle, \text{ where } \mathrm{v} \in \mathcal{F}$$

For an operator $\mathrm{D} \in HS(\mathcal{F})$ with rank $k$ with spectral decomposition $\mathrm{D} = \sum_{i=1}^{\infty} \mu_i \psi_i \otimes \psi_i$, let $\Psi$ be the matrix with eigenvectors $\psi_i$ as columns and $\Phi$ be the matrix with eigenvectors of $\mathrm{C}_m$ as columns (see Definition 3.4). Define

$$\mathrm{R}^* = \underset{\mathrm{R}^\top \mathrm{R} = \mathrm{R} \mathrm{R}^\top = \mathrm{I}}{\arg \min} \|\Psi \mathrm{R} - \Phi\|_{\mathcal{F}}^2, \quad \tilde{\Psi} := \Psi \mathrm{R}^*$$

Let $\tilde{\psi}_i$ be the columns of $\tilde{\Psi}$, define

$$\mathfrak{A} : HS(\mathcal{F}) \to HS(\rho), \ \mathfrak{A}\mathrm{D} := \sum_{i=1}^{k} \mu_i \frac{\mathrm{A}\tilde{\psi}_i}{\sqrt{\langle \mathrm{C}_m \tilde{\psi}_i, \tilde{\psi}_i \rangle_{\mathcal{F}}}} \otimes_\rho \frac{\mathrm{A}\tilde{\psi}_i}{\sqrt{\langle \mathrm{C}_m \tilde{\psi}_i, \tilde{\psi}_i \rangle_{\mathcal{F}}}}$$

Note that the definition of the approximation operator $\mathfrak{A}$ requires knowledge of the co-variance matrix $\mathrm{C}_m$ to find the optimal rotation matrix $\mathrm{R}^*$, but this is solely for the purpose of analysis and is not used in the algorithm in any form.

In Proposition A.11, we show that the adjoint of the Approximation Operator is

$$\mathrm{A}^* : L^2(\mathrm{X}, \rho) \to \mathcal{F}, (\mathrm{A}^* f)_i = \int_{\mathcal{X}} f(\mathrm{x}) z_{\omega_i}(\mathrm{x}) d\rho(\mathrm{x}).$$

Moreover, in Proposition A.12 we show that that the approximate covariance operator and the approximate kernel integral operator can be expressed in terms of the Approximation operator A as $\mathrm{C}_m = \mathrm{A}^* \mathrm{A}$ and $\mathrm{L}_m = \mathrm{A} \mathrm{A}^*$.

**Proposition A.11.** *The approximation operator satisfies the following properties,*

*(a). The adjoint of* A *is* $(\mathrm{A}^* f)_i = \frac{1}{\sqrt{m}} \int_{\mathcal{X}} f(\mathrm{x}) z_{\omega_i}(\mathrm{x}) d\rho(\mathrm{x})$

*(b).* A *and* $\mathrm{A}^*$ *are Hilbert-Schmidt.*

*Proof of Proposition A.11. (a).* First we show that the adjoint of A is $(\mathrm{A}^* f)_i = \frac{1}{\sqrt{m}} \int_{\mathcal{X}} f(\mathrm{x}) z_{\omega_i}(\mathrm{x}) d\rho(\mathrm{x})$. For $\mathrm{v} \in \mathcal{F}, f \in L^2(\mathcal{X}, \rho)$, we have,

$$
\begin{aligned}
\langle \mathrm{A}\mathrm{v}, f \rangle_\rho &= \int_{\mathcal{X}} (\mathrm{A}\mathrm{v})(\mathrm{x}) f(\mathrm{x}) d\rho(\mathrm{x}) \\
&= \int_{\mathcal{X}} \langle z(\mathrm{x}), \mathrm{v} \rangle_{\mathcal{F}} f(\mathrm{x}) d\rho(\mathrm{x}) && \text{(Definition A.10)} \\
&= \left\langle \int_{\mathcal{X}} z(\mathrm{x}) f(\mathrm{x}) d\rho(\mathrm{x}), \mathrm{v} \right\rangle_{\mathcal{F}} && \text{(Fubini's Theorem)} \\
&= \langle \mathrm{A}^* f, \mathrm{v} \rangle_{\mathcal{F}}
\end{aligned}
$$

*(b).* Let $\{\mathrm{e}_i\}$ be an orthonormal basis for $\mathcal{F}$.

$$
\begin{aligned}
\|\mathrm{A}\|_{\mathcal{L}(\mathcal{F}, \rho)}^2 &= \sum_{i=1}^{m} \|\mathrm{A}\mathrm{e}_i\|_\rho^2 && \text{(Pythagoras Theorem)} \\
&= \sum_{i=1}^{m} \|\langle z(\cdot), \mathrm{e}_i \rangle_{\mathcal{F}}\|_\rho^2 && \text{(Definition 3.6)} \\
&= \sum_{i=1}^{m} \int_{\mathcal{X}} (\langle z(\mathrm{x}), \mathrm{e}_i \rangle_{\mathcal{F}})^2 d\rho(\mathrm{x}) \\
&\leq \sum_{i=1}^{m} \int_{\mathcal{X}} \|z(\mathrm{x})\|_{\mathcal{F}}^2 d\rho(\mathrm{x}) < m\tau^2 < \infty
\end{aligned}
$$

where third last and second inequality follows from Cauchy Schwartz inequality and Assumption 3.1 respectively.

Similarly, to show $\mathrm{A}^*$ is Hilbert-Schmidt, we note that $\|\mathrm{A}^*\|_{\mathcal{L}^2(\rho, \mathcal{F})} = \|\mathrm{A}\|_{\mathcal{L}^2(\mathcal{F}, \rho)}^2 < \infty$ $\qquad \square$

In the following proposition, we show how the Covariance operator $C_m$ and kernel Integral operator $L_m$ are related.

**Proposition A.12.** *The following properties hold,*

   (a). *$C_m$ and $L_m$ satisfy that $C_m = A^*A, L_m = AA^*$*

   (b). *$C_m$ and $L_m$ are trace-class.*

*Proof of Proposition A.12. (a).* We first show the first part of the Proposition. For any $v \in \mathcal{F}$, we have,

$$
\begin{aligned}
A^*Av &= \int_{\mathcal{X}} \langle z(x), v \rangle_{\mathcal{F}} z(x) d\rho(x) && \text{(Definition A.10 and Proposition A.11)} \\
&= \mathbb{E}_{\rho} \left[ z(x) \otimes_{\mathcal{F}} z(x) \right] v \\
&= C_m v && \text{(Definition A.2)}
\end{aligned}
$$

For any $g \in L^2(\mathcal{X}, \rho)$,

$$
\begin{aligned}
AA^*g &= \frac{1}{m} \sum_{i=1}^m z_{\omega_i}(\cdot) \int_{\mathcal{X}} z_{\omega_i}(x) g(x) d\rho(x) && \text{(Definition A.10 and Proposition A.11)} \\
&= \int_{\mathcal{X}} \sum_{i=1}^m \frac{1}{\sqrt{m}} z_{\omega_i}(\cdot) \frac{1}{\sqrt{m}} z_{\omega_i}(x) g(x) d\rho(x) && \text{(Fubini's Theorem)} \\
&= \int_{\mathcal{X}} \langle z(x), z(\cdot) \rangle_{\mathcal{F}} g(x) d\rho(x) \\
&= \int_{\mathcal{X}} k_m(x, \cdot) g(x) d\rho(x, t) && \text{(Definition of the approximate kernel mapping)} \\
&= L_m g && \text{(Definition A.4)}
\end{aligned}
$$

*(b).* Now we show that $C_m$ and $L_m$ are trace-class.

$$
\begin{aligned}
\|C_m\|_{\mathcal{L}^1(\mathcal{F})} &= \|A^*A\|_{\mathcal{L}^1(\mathcal{F})} && \text{(Proposition A.12)} \\
&= \|A\|_{\mathcal{L}^2(\mathcal{F})}^2 < \infty && \text{(Proposition A.11)}
\end{aligned}
$$

Similarly,

$$
\begin{aligned}
\|L_m\|_{\mathcal{L}^1(\rho)} &= \|AA^*\|_{\mathcal{L}^1(\rho)} && \text{(Proposition A.12)} \\
&= \|A^*\|_{\mathcal{L}^2(\rho)}^2 < \infty && \text{(Proposition A.11)}
\end{aligned}
$$

$\square$

## A.4    Kernel integral operator $L$ and its approximation $L_m$

We first recall the definition of Kernel integral operator $L$ and its approximation $L_m$.

**Definition A.13.** The kernel integral operator $L : L^2(\mathcal{X}, \rho) \to L^2(\mathcal{X}, \rho)$ is defined as follows:

$$
Lg = \int_{\mathcal{X}} k(x, \cdot) g(x) d\rho(x) \ \forall \ g \in L^2(\mathcal{X}, \rho)
$$

**Definition A.14.** $L_m : L^2(\mathcal{X}, \rho) \to L^2(\mathcal{X}, \rho)$ is the (approximated) kernel integral operator, defined as:

$$
(L_m g)(\cdot) = \int_{\mathcal{X}} k_m(x, \cdot) g(x) d\rho(x)
$$

We now show in Proposition A.15 that spectral decomposition of the kernel integral operator $L$ can be given in terms of the eigenfunctions and the eigenvalues of the covariance operator $C$.

Figure 3: Maps between the data domain ($\mathcal{X}$), space of square integrable functions on $\mathcal{X}$ ($L^2(\mathcal{X}, \rho)$), the RKHS of kernel $k(\cdot, \cdot)$, and RKHS of the approximate feature map, as well as maps between Hilbert-Schmidt operators on these spaces.

**Proposition A.15.** *The spectral decomposition of* L *is:*

$$L = \sum_{i=1}^{\infty} \bar{\lambda}_i \frac{I\bar{\phi}_i}{\sqrt{\bar{\lambda}_i}} \otimes_\rho \frac{I\bar{\phi}_i}{\sqrt{\bar{\lambda}_i}}$$

*where* $\frac{I\bar{\phi}_i}{\sqrt{\bar{\lambda}_i}}$ *are the (unit norm) eigenfunctions of* L *with eigenvalues* $\bar{\lambda}_i$

*Proof of Proposition A.15.* First we show that the operators L and $\sum_{i=1}^{\infty} \bar{\lambda}_i \frac{I\bar{\phi}_i}{\sqrt{\bar{\lambda}_i}} \otimes_\rho \frac{I\bar{\phi}_i}{\sqrt{\bar{\lambda}_i}}$ agree on any function $g \in L^2(\mathcal{X}, \rho)$.

$$\sum_{i=1}^{\infty} \bar{\lambda}_i \frac{I\bar{\phi}_i}{\sqrt{\bar{\lambda}_i}} \otimes_\rho \frac{I\bar{\phi}_i}{\sqrt{\bar{\lambda}_i}} g = \sum_{i=1}^{\infty} I\bar{\phi}_i \left\langle I\bar{\phi}_i, g \right\rangle_\rho \qquad \text{(Definition of outer product)}$$

$$= \sum_{i=1}^{\infty} \bar{\phi}_i \left\langle \bar{\phi}_i, g \right\rangle_\rho \qquad \text{(Definition A.7)}$$

$$= \sum_{i=1}^{\infty} \bar{\phi}_i \int_{\mathcal{X}} \left\langle \bar{\phi}_i, k(\mathrm{x}, \cdot) \right\rangle_{\mathcal{H}} g(\mathrm{x}, t) d\rho(\mathrm{x}, t) \qquad \text{(Reproducing property)}$$

$$= \sum_{i=1}^{\infty} \bar{\phi}_i \left\langle \bar{\phi}_i, \int_{\mathcal{X}} k(\mathrm{x}, \cdot) g(\mathrm{x}, t) d\rho(\mathrm{x}, t) \right\rangle_{\mathcal{H}} \qquad \text{(Fubini's Theorem)}$$

$$= \int_{\mathcal{X}} k(\mathrm{x}, \cdot) g(\mathrm{x}, t) d\rho(\mathrm{x}, t) \qquad \text{(Pythagoras theorem)}$$

$$= Lg$$

where the second to last equality holds by Pythagoras theorem, since $\bar{\phi}_i$'s are basis for $\mathcal{H}$. Now we show that $\frac{I\bar{\phi}_i}{\sqrt{\bar{\lambda}_i}}$ are eigenfunctions of L with eigenvalues $\bar{\lambda}_i$.

$$\mathrm{L}\frac{\mathrm{I}\bar{\phi}_i}{\sqrt{\bar{\lambda}_i}} = \mathrm{II}^* \frac{\mathrm{I}\bar{\phi}_i}{\sqrt{\bar{\lambda}_i}} \qquad\qquad \text{(Proposition A.9)}$$

$$= \mathrm{IC}\frac{\bar{\phi}_i}{\sqrt{\bar{\lambda}_i}} \qquad\qquad \text{(Proposition A.9)}$$

$$= \bar{\lambda}_i \frac{\mathrm{I}\bar{\phi}_i}{\sqrt{\bar{\lambda}_i}} \qquad\qquad (\bar{\phi}_i \text{ is an eigenfunction of C})$$

Moreover, they have unit norms and are mutually orthogonal.

$$\left\langle \frac{\mathrm{I}\bar{\phi}_i}{\sqrt{\bar{\lambda}_i}}, \frac{\mathrm{I}\bar{\phi}_j}{\sqrt{\bar{\lambda}_j}} \right\rangle_\rho = \frac{1}{\sqrt{\bar{\lambda}_i \bar{\lambda}_j}} \left\langle \mathrm{I}^* \mathrm{I}\bar{\phi}_i, \bar{\phi}_j \right\rangle_\mathcal{H} \qquad \text{(Proposition A.8)}$$

$$= \frac{1}{\sqrt{\bar{\lambda}_i \bar{\lambda}_j}} \left\langle \mathrm{C}\bar{\phi}_i, \bar{\phi}_j \right\rangle_\mathcal{H} \qquad \text{(Proposition A.9)}$$

$$= \frac{1}{\sqrt{\bar{\lambda}_i \bar{\lambda}_j}} \lambda_i \delta_{ij} = \delta_{ij}$$

$$\square$$

Similarly, we characterize the relationship between the spectral decomposition of the approximate kernel integral operator $\mathrm{L}_m$ and the approximate covariance operator $\mathrm{C}_m$ in Proposition A.16.

**Proposition A.16.** *The spectral decomposition of* $\mathrm{L}_m$ *is:*

$$\mathrm{L}_m = \sum_{i=1}^m \lambda_i \frac{\mathrm{A}\phi_i}{\sqrt{\lambda_i}} \otimes_\rho \frac{\mathrm{A}\phi_i}{\sqrt{\lambda_i}}$$

*where* $\frac{\mathrm{A}\phi_i}{\sqrt{\lambda_i}}$ *are the (unit norm) eigenfunctions of* $\mathrm{L}_m$ *with eigenvalues* $\lambda_i$

*Proof of Proposition A.16.* First we prove that $\mathrm{L}_m$ and $\sum_{i=1}^m \lambda_i \frac{\mathrm{A}\phi_i}{\sqrt{\lambda_i}} \otimes_\rho \frac{\mathrm{A}\phi_i}{\sqrt{\lambda_i}}$ agree on any function $g$ in $L^2(\mathcal{X}, \rho)$:

$$\sum_{i=1}^m \lambda_i \frac{\mathrm{A}\phi_i}{\sqrt{\lambda_i}} \otimes_\rho \frac{\mathrm{A}\phi_i}{\sqrt{\lambda_i}} g = \sum_{i=1}^m \langle A\phi_i, g \rangle_\rho A\phi_i \qquad\qquad \text{(Definition of outer product)}$$

$$= \sum_{i=1}^m \int_\mathcal{X} \langle \mathrm{z}(\mathrm{x},t), \phi_i \rangle_\mathcal{F} g(\mathrm{x},t) d\rho(\mathrm{x},t) \langle \mathrm{z}(\cdot), \phi_i \rangle_\mathcal{F} \qquad \text{(Definition A.10)}$$

$$= \int_\mathcal{X} \sum_{i=1}^m \langle \mathrm{z}(\mathrm{x},t), \phi_i \rangle_\mathcal{F} \langle \mathrm{z}(\cdot), \phi_i \rangle_\mathcal{F} g(\mathrm{x},t) d\rho(\mathrm{x},t) \qquad \text{(Fubini's Theorem)}$$

$$= \int_\mathcal{X} \sum_{i=1}^m \frac{1}{\sqrt{m}} z_{\omega_i}(\mathrm{x},t) \frac{1}{\sqrt{m}} z_{\omega_i}(\cdot) g(\mathrm{x},t) d\rho(\mathrm{x},t) \qquad \text{(Change of basis)}$$

$$= \int_\mathcal{X} \langle \mathrm{z}(\mathrm{x},t), \mathrm{z}(\cdot) \rangle_\mathcal{F} g(\mathrm{x},t) d\rho(\mathrm{x},t)$$

$$= \int_\mathcal{X} k_m(\mathrm{x},\cdot) g(\mathrm{x},t) d\rho(\mathrm{x},t)$$

$$= \mathrm{L}_m g$$

The fourth equality follows from the fact that inner products are invariant under orthogonal change of basis. Now we show that $\frac{A\phi_i}{\sqrt{\lambda_i}}$ are eigenfunctions of $L_m$ with the corresponding eigenvalue $\lambda_i$

$$L_m \frac{A\phi_i}{\lambda_i} = AA^* \frac{A\phi_i}{\sqrt{\lambda_i}} \qquad \text{(Proposition A.12)}$$

$$= \frac{AC_m\phi_i}{\sqrt{\lambda_i}} \qquad \text{(Proposition A.12)}$$

$$= \lambda_i \frac{A\phi_i}{\sqrt{\lambda_i}} \qquad (\phi_i \text{ is an eigenvector of } C_m)$$

Moreover, they have unit norms and are mutually orthogonal.

$$\left\langle \frac{A\phi_i}{\sqrt{\lambda_i}}, \frac{A\phi_j}{\sqrt{\lambda_j}} \right\rangle_\rho = \frac{1}{\sqrt{\lambda_i\lambda_j}} \langle A^*A\phi_i, \phi_j \rangle_{\mathcal{F}} \qquad \text{(Proposition A.11)}$$

$$= \frac{1}{\sqrt{\lambda_i\lambda_j}} \langle C_m\phi_i, \phi_j \rangle_{\mathcal{F}} \qquad \text{(Proposition A.12)}$$

$$= \frac{1}{\sqrt{\lambda_i\lambda_j}} \lambda_i \delta_{ij} = \delta_{ij}$$

which completes the proof. $\qquad\qquad\square$

The following is an important lemma which shows that the kernel integral operator and its approximation can be seen as true and empirical covariance operators in $HS(\rho)$ associated with the random variable $z_\omega$. This allows us to use concentration of measure tools to bound the approximation error in $H(\rho)$.

**Lemma A.17.** $L = \mathbb{E}_\omega \left[ z_\omega \otimes_\rho z_\omega \right], L_m = \frac{1}{m} \sum_{i=1}^m z_{\omega_i} \otimes_\rho z_{\omega_i}$

*Proof of Lemma A.17.* For any $f, g \in L^2(\mathcal{X}, \rho)$ it holds that

$$\langle Lf, g \rangle_\rho = \left\langle \int_{\mathcal{X}} k(x, \cdot) f(x, t) d\rho(x, t), g \right\rangle_\rho \qquad \text{(Definition A.3)}$$

$$= \left\langle \int_{\mathcal{X}} \int_{\Omega} z_\omega(x, t) z_\omega(\cdot) f(x, t) d\pi(\omega) d\rho(x, t), g \right\rangle_\rho \qquad \text{(Theorem A.6)}$$

$$= \left\langle \int_{\Omega} \int_{\mathcal{X}} z_\omega(x, t) f(x, t) d\rho(x, t) z_\omega(\cdot) d\pi(\omega), g \right\rangle_\rho \qquad \text{(Fubini's theorem)}$$

$$= \left\langle \int_{\Omega} \langle z_\omega(\cdot), f \rangle_\rho z_\omega(\cdot) d\pi(\omega), g \right\rangle_\rho$$

$$= \left\langle \int_{\Omega} z_\omega(\cdot) \otimes_\rho z_\omega(\cdot) d\pi(\omega) f, g \right\rangle_\rho \qquad \text{(Definition of outer product)}$$

$$= \langle \mathbb{E}_\omega \left[ z_\omega \otimes_\rho z_\omega \right] f, g \rangle_\rho$$

Similarly, for any $f, g \in L^2(\mathcal{X}, \rho)$ we have that

$$
\begin{aligned}
\langle \mathrm{L}_m f, g \rangle_\rho &= \left\langle \int_{\mathcal{X}} k_m(\mathrm{x}, \cdot) f(\mathrm{x}, t) d\rho(\mathrm{x}, t), g \right\rangle_\rho && \text{(Definition A.4)} \\
&= \left\langle \int_{\mathcal{X}} \langle \mathrm{z}(\mathrm{x}, t), \mathrm{z}(\cdot) \rangle_{\mathcal{F}} f(\mathrm{x}, t) d\rho(\mathrm{x}, t), g \right\rangle_\rho && \text{(Definition of } k_m) \\
&= \left\langle \int_{\mathcal{X}} \frac{1}{m} \sum_{i=1}^m z_{\omega_i}(\mathrm{x}, t) z_{\omega_i}(\cdot) f(\mathrm{x}, t) d\rho(\mathrm{x}, t), g \right\rangle_\rho \\
&= \left\langle \frac{1}{m} \sum_{i=1}^m \int_{\mathcal{X}} z_{\omega_i}(\mathrm{x}, t) f(\mathrm{x}, t) d\rho(\mathrm{x}, t) z_{\omega_i}(\cdot), g \right\rangle_\rho \\
&= \left\langle \frac{1}{m} \sum_{i=1}^m \langle z_{\omega_i}(\cdot), f \rangle_\rho \, z_{\omega_i}(\cdot), g \right\rangle_\rho \\
&= \left\langle \frac{1}{m} \sum_{i=1}^m z_{\omega_i}(\cdot) \otimes_\rho z_{\omega_i}(\cdot) f, g \right\rangle_\rho && \text{(Definition of outer product)} \\
&= \langle \mathrm{L}_m f, g \rangle_\rho && \text{(Definition A.4)}
\end{aligned}
$$

which completes the proof. $\qquad\square$

The following Proposition shows the relation between the outer products in two separable Hilbert spaces; this is useful in the proof of the main theorem.

**Proposition A.18.** *For any Hilbert-Schmidt Operator* $\mathrm{B} : \mathcal{H}_1 \to \mathcal{H}_2$, *it holds that* $\mathrm{B}u \otimes_{\mathcal{H}_2} \mathrm{B}v = \mathrm{B}(u \otimes_{\mathcal{H}_1} v) \mathrm{B}^*$, *where* $\mathrm{u}, \mathrm{v} \in \mathcal{H}_1$.

*Proof of Proposition A.18.* For any $f \in \mathcal{H}_2$, the following equalities hold:

$$
\begin{aligned}
(\mathrm{B}u \otimes_{\mathcal{H}_2} \mathrm{B}v) f &= \mathrm{B}u \langle \mathrm{B}v, f \rangle_{\mathcal{H}_2} && \text{(Definition of outer product)} \\
&= \mathrm{B}u \langle v, \mathrm{B}^* f \rangle_{\mathcal{H}_1} && \text{(Proposition A.11)} \\
&= \mathrm{B}(u \langle v, \mathrm{B}^* f \rangle_{\mathcal{H}_1}) \\
&= \mathrm{B}(u \otimes_{\mathcal{H}_1} v) \mathrm{B}^* f && \text{(Definition of outer product)}
\end{aligned}
$$

$\qquad\square$

Finally, we state the assumptions that we make throughout the paper.

**Assumption A.19.** The kernel function $k$ is a Mercer's kernel(see Theorem A.5) and has the following integral representation, $k(\mathrm{x}, \mathrm{y}) = \int_\Omega z(\mathrm{x}, \omega) z(\mathrm{y}, \omega) d\pi(\omega) \; \forall \mathrm{x}, \mathrm{y} \in \mathcal{X}$ where $(\mathcal{H}, k)$ is a separable RKHS of real-valued functions on $\mathcal{X}$ with a bounded positive definite kernel $k$. We also assume that there exists $\tau > 1$ such that $|z(\mathrm{x}, \omega)| \leq \tau$ for all $x \in \mathcal{X}, \omega \in \Omega$. Furthermore, we assume that the operator $\mathrm{L}^{\frac{1}{2}}$ exists.

# B   Equivalence of optimization problems in $\mathcal{H}$ and $L^2(\mathcal{X}, \rho)$

We now show that Kernel PCA in the RKHS $\mathcal{H}$ and $L^2(\mathcal{X}, \rho)$ are equivalent under some assumptions which we show are naturally satisfied in the case of Kernel PCA with random features. Ths allows us to transfer our generalization bounds established in $L^2(\mathcal{X}, \rho)$ to $\mathcal{H}$.

The Kernel PCA problem essentially reduces to solving the following optimization problem:

$$\max_{\mathrm{P} \in \mathcal{P}_{\mathcal{H}}^k} \langle \mathrm{P}, \mathrm{C} \rangle_{HS(\mathcal{H})} \tag{OPT-1}$$

For any $\mathrm{P} \in \mathcal{P}_{HS(\mathcal{H})}^k$, by spectral decomposition, $\mathrm{P}$ has an eigendecompostion given by $\mathrm{P} = \sum_{i=1}^k \mathrm{u}_i \otimes_\rho \mathrm{u}_i$ where $\mathrm{u}_i \in \mathcal{H}, i \in [k]$ are a set of orthonormal functions. We define an operator $\mathrm{U} : \mathbb{R}^k \to \mathcal{H}$ such that $\mathrm{Ub} = \sum_{i=1}^k \mathrm{b}_i \mathrm{u}_i$, where $\mathrm{b} \in \mathbb{R}^k$.

**Proposition B.1.** $\mathrm{U}$ *satisfies the following properties.*

*(a)* $\mathrm{U}$ *is Hilbert-Schmidt.*

*(b)* *The adjoint of* $\mathrm{U}$ *is* $\mathrm{U}^* : \mathcal{H} \to \mathbb{R}^k$ *such that* $(\mathrm{U}^* f)_i = \langle \mathrm{u}_i, f \rangle_{\mathcal{H}}$ *where* $f \in \mathcal{H}$.

*(c)* $\mathrm{P} = \mathrm{U}\mathrm{U}^*$ *and* $\mathrm{U}^*\mathrm{U} = \mathrm{I}_k$

*Proof. (a)* First we show that the operator $\mathrm{U}$ is Hilbert-Schmidt. Let $\{\mathrm{e}_i\}_{i=1}^k$ be the canonical basis of $\mathbb{R}^k$.

$$\|\mathrm{U}\|_{\mathcal{L}^2(\mathbb{R}^k, \mathcal{H})}^2 = \sum_{i=1}^k \|\mathrm{Ue}_i\|_{\mathcal{H}}^2 \qquad \text{(Pythagoras Theorem)}$$

$$= \sum_{i=1}^k \|\mathrm{u}_i\|_{\mathcal{H}}^2 = k$$

*(b)* Let $\mathrm{U}^*$ be the adjoint of $\mathrm{U}$. We now show that $(\mathrm{U}^* f)_i = \langle \mathrm{u}_i, f \rangle_{\mathcal{H}}$. For any $\mathrm{b} \in \mathbb{R}^k, f \in \mathcal{H}$,

$$\langle \mathrm{U}^* f, \mathrm{b} \rangle = \langle f, \mathrm{Ub} \rangle_{\mathcal{H}}$$

$$= \left\langle f, \sum_{i=1}^k \mathrm{b}_i \mathrm{u}_i \right\rangle_{\mathcal{H}}$$

$$= \sum_{i=1}^k \langle f, \mathrm{u}_i \rangle_{\mathcal{H}} \, \mathrm{b}_i$$

$$= \langle \mathrm{d}, \mathrm{b} \rangle$$

where $\mathrm{d} \in \mathbb{R}^k, \mathrm{d}_i = \langle f, \mathrm{u}_i \rangle_{\mathcal{H}}$

*(c)* For the first part, for any $f \in \mathcal{H}$, we have,

$$\mathrm{P}f = \sum_{i=1}^k (\mathrm{u}_i \otimes_{\mathcal{H}} \mathrm{u}_i) f$$

$$= \sum_{i=1}^k \langle \mathrm{u}_i, f \rangle_{\mathcal{H}} \, \mathrm{u}_i$$

$$= \sum_{i=1}^k (\mathrm{U}^* f)_i \mathrm{u}_i$$

$$= \mathrm{U}\mathrm{U}^* f$$

Now we show that the constraint $\mathrm{P} \in \mathcal{P}_{\mathcal{H}}^k$ reduces to $\mathrm{U}^*\mathrm{U} = \mathrm{I}_k$.

For any $b \in \mathbb{R}^k$,

$$U^*Ub = \sum_{i=1}^{k} b_i U^* u_i$$

$$= \sum_{i=1}^{k} b_i d_i$$

where $d_i \in \mathbb{R}^k$, $(d_i)_j = \langle u_j, u_i \rangle_{\mathcal{H}}$. Note that since $u_i$'s are orthonormal functions, therefore, $d_i = e_i$, where $e_i$'s is the canonical basis of $\mathbb{R}^k$. Therefore,

$$U^*Ub = \sum_{i=1}^{k} b_i e_i = b$$

$\square$

We now write the optimization problem OPT-1 in terms of U as,

$$\max_{U^*U=I_k} \langle UU^*, C \rangle_{HS(\mathcal{H})} \tag{OPT-2}$$

Now consider $v_i \in L^2(\mathcal{X}, \rho), i \in [k]$ such that $u_i = I^* v_i$. Note that the existence of $v_i$ is guaranteed from the construction of RKHS from eigenfunctions of L (for details, see [Sejdinovic and Gretton, 2012, Theorem 51]). We now define an operator $V : \mathbb{R}^k \to L^2(\mathcal{X}, \rho)$ such that $Vb = \sum_{i=1}^{k} b_i v_i$, where $b \in \mathbb{R}^k$. We have the following proposition about V.

**Proposition B.2.** V *satisfies the following properties,*

*(a)* V *is Hilbert-Schmidt.*

*(b) The adjoint of* V *is* $V^* : L^2(\mathcal{X}, \rho) \to \mathcal{H}$, *defined as* $(V^* f)_i = \langle v_i, f \rangle_\rho$

*(c)* $\langle VV^*, L^2 \rangle_{HS(\rho)} = \langle UU^*, C \rangle_{HS(\mathcal{H})}$ *and* $V^*LV = U^*U = I_k$

*Proof.* The proofs of *(a)* and *(b)* are similar to that of Proposition B.1.
*(c).* We start with the first part. The objective in terms of V is,

$$\langle P, C \rangle_{HS(\mathcal{H})} = \left\langle \sum_{i=1}^{k} u_i \otimes_{\mathcal{H}} u_i, C \right\rangle_{HS(\mathcal{H})}$$

$$= \left\langle \sum_{i=1}^{k} I^* v_i \otimes_{\mathcal{H}} I^* v_i, C \right\rangle_{HS(\mathcal{H})} \qquad (u_i = I^* v_i)$$

$$= \left\langle \sum_{i=1}^{k} I(v_i \otimes_\rho v_i) I^*, C \right\rangle_{HS(\mathcal{H})} \qquad \text{(Proposition A.18)}$$

$$= \left\langle \sum_{i=1}^{k} v_i \otimes_\rho v_i, ICI^* \right\rangle_{HS(\rho)} \qquad \text{(Definition of adjoint)}$$

$$= \left\langle \sum_{i=1}^{k} v_i \otimes_\rho v_i, II^*II^* \right\rangle_{HS(\rho)} \qquad \text{(Proposition A.9)}$$

$$= \left\langle \sum_{i=1}^{k} v_i \otimes_\rho v_i, L^2 \right\rangle_{HS(\rho)} \qquad \text{(Proposition A.9)}$$

$$= \langle VV^*, L^2 \rangle_{HS(\rho)}$$

For the second part, for any $b \in \mathbb{R}^k$, we have,

$$U^*Ub = \sum_{i=1}^{k} b_i U^*u_i$$

$$= \sum_{i=1}^{k} b_i U^*I^*v_i \qquad\qquad (u_i = I^*v_i)$$

$$= \sum_{i=1}^{k} b_i d_i$$

where $d_i \in \mathbb{R}^k$, $(d_i)_j = \langle u_j, I^*v_i \rangle_{\mathcal{H}} = \langle I^*v_j, I^*v_i \rangle_{\mathcal{H}} = \langle v_j, II^*v_i \rangle_{\rho} = \langle v_j, Lv_i \rangle_{\rho}$, where the third equality follows from the property of adjoints, and the last equality because $L = II^*$.
Since $U^*Ub = b$, so $d_i = e_i$. Therefore, we get $\langle v_j, Lv_i \rangle = \delta_{ij}$

Let us now look at the $j^{\text{th}}$ element of $V^*LVb$,

$$(V^*LVb)_j = (V^*L\sum_{i=1}^{k} b_i v_i)_j \qquad\qquad \text{(Definition of V)}$$

$$= \sum_{i=1}^{k} (b_i V^*Lv_i)_j$$

$$= \sum_{i=1}^{k} b_i \langle v_j, Lv_i \rangle_{\rho}$$

$$= \sum_{i=1}^{k} b_i \delta_{ij} = b_j$$

$$\square$$

We can now restate the optimization problem in terms of V.

$$\max_{V^*LV=I_k} \langle VV^*, L^2 \rangle_{HS(\rho)} \qquad\qquad \text{(OPT-3)}$$

Now, let $w_i = L^{1/2}v_i$. Note that $w_i$ is well-defined since we assume that $L^{1/2}$ exists (See Assumption 3.1). Define $W : \mathbb{R}^k \to L^2(\mathcal{X}, \rho)$, such that $Wb = \sum_{i=1}^{k} b_i w_i$.

**Proposition B.3.** $W$ *satisfies the following properties,*

*(a)* $W$ *is Hilbert-Schmidt.*

*(b) The adjoint of* $W$ *is* $W^* : L^2(\mathcal{X}, \rho) \to \mathbb{R}^k$ $(W^*f)_i = \langle w_i, f \rangle_{\rho}$.

*(c)* $W = L^{1/2}V$, $\langle VV^*, L^2 \rangle_{HS(\rho)} = \langle WW^*, L \rangle_{HS(\rho)}$ *and* $W^*W = V^*LV = I_k$

*Proof.* The proofs of *(a)* and *(b)* are similar to that of Proposition B.1.
*(c)* For the first part, for any $b \in \mathbb{R}^k$, we have

$$Wb = \sum_{i=1}^{k} b_i w_i$$

$$= \sum_{i=1}^{k} b_i L^{1/2}v_i$$

$$= L^{1/2}\sum_{i=1}^{k} b_i v_i$$

$$= L^{1/2}Vb$$

Note that since L is self-ajoint, $L^{1/2}$ is self-adjoint too. The objective in terms of W is

$$\langle VV^*, L^2 \rangle_{HS(\rho)} = \left\langle L^{1/2}VV^*L^{1/2}, L \right\rangle_{HS(\rho)} \qquad \text{(Definition of adjoint)}$$

$$= \langle WW^*, L \rangle_{HS(\rho)}$$

Equivalently, we can restate the constraint in terms of W as,

$$V^*LV = V^*L^{1/2}L^{1/2}V$$

$$= (L^{1/2}V)^*(L^{1/2}V)$$

$$= W^*W = I_k$$

$\square$

We now restate the optimization problem in terms of W.

$$\max_{W^*W=I_k} \langle WW^*, L \rangle_{HS(\rho)} \qquad \text{(OPT-4)}$$

We now state this equivalence of objective in the following Lemma.

**Lemma B.4** (Equivalence of Objective)**.**

$$\langle P, C \rangle_{HS(\mathcal{H})} = \langle WW^*, L \rangle_{HS(\rho)}$$

*where the relation between* P *and* W *is presented via Propositions B.1,B.2 and B.3.*

*Proof.* One direction of implication simply simply follows from the construction in Propositions B.1, B.2 and B.3. In particular, from Propositions B.1, B.2 and B.3, we conclude that OPT-1 $\implies$ OPT-2 $\implies$ OPT-3 $\implies$ OPT-4. It is easy to see that OPT-3 $\implies$ OPT-2 $\implies$ OPT-1 where the first implication simply follows from the construction of $u_i$'s and the second from Proposition B.1. However, showing that OPT-4 $\implies$ OPT-3 is conditioned on $w_i$'s lying in the range of $L^{1/2}$ because otherwise there might not exist $v_i$'s such that $w_i = L^{1/2}v_i$. In Lemma B.5, we show that when using random features, with the approximation operator defined in Definition 3.6, the functions obtained via random feature approximation lies in the range of $L^{1/2}$ with probability 1 on the support of $\pi$. This establishes the equivalence claimed. $\square$

We now formally show that vectors from $\mathcal{F}$ lifted to $L^2(\mathcal{X}, \rho)$ via the approximation operator A lie in the range of $L^{1/2}$ almost surely with respect to measure $\pi$. The proof of the following lemma closely follows [Rudi and Rosasco, 2017, Lemma 2].

**Lemma B.5.** *For every* $v \in \mathcal{F}$, $Av \in L^2(\mathcal{X}, \rho)$ *lies in the range of* $L^{1/2}$ *almost surely on the support of* $\pi$.

*Proof.* Let $\Pi \in HS(\rho)$ denote the projection operator projecting to the range of $L^{1/2}$. Then $(I_\rho - \Pi)L^{1/2}f = 0 \ \forall \ f \in L^2(\mathcal{X}, \rho)$ as $(I_\rho - \Pi)$ is the projection to the orthogonal complement to the range of $L^{1/2}$. From this, we have, $\text{Tr}\left((I_\rho - \Pi)L^{1/2}L^{1/2}(I_\rho - \Pi)\right) = \text{Tr}\left((I_\rho - \Pi)L(I_\rho - \Pi)\right) = 0$.

$$\text{Tr}\left((I_\rho - \Pi)L(I_\rho - \Pi)\right) = \text{Tr}\left((I_\rho - \Pi)\int_\Omega z_\omega \otimes_\rho z_\omega d\pi(\omega)(I_\rho - \Pi)\right)$$

$$= \int_\Omega \text{Tr}\left((I_\rho - \Pi)(z_\omega \otimes_\rho z_\omega)(I_\rho - \Pi)\right) d\pi(\omega)$$

$$= \int_\Omega \text{Tr}\left((I_\rho - \Pi)z_\omega \otimes_\rho (I_\rho - \Pi)z_\omega\right) d\pi(\omega)$$

$$= \int_\Omega \|(I_\rho - \Pi)z_\omega\|_\rho^2 d\pi(\omega) = 0$$

From the above equation, we see that $\|(I_\rho - \Pi)z_\omega\|_\rho = 0$ almost surely on the support of $\pi$. This implies that $(I_\rho - \Pi)z_\omega = 0$ a.s. on the support of $\pi$.

Now we show that all functions of interest i.e. anything lifted from $\mathcal{F}$ to $L^2(\mathcal{X}, \rho)$ lie in the range of $L^{1/2}$. Let $v \in \mathcal{F}, f \in L^2(\mathcal{X}, \rho)$.

$$\langle (I_\rho - \Pi)f, Av \rangle_\rho = \langle A^*(I_\rho - \Pi)f, v \rangle_\mathcal{F}$$
$$= \sum_{i=1}^{m} ((I_\rho - \Pi)z_{\omega_i}f)v_i = 0$$

This is because $\omega_i$'s are drawn from $\pi$, and we already argued that $(I_\rho - \Pi)z_\omega = 0$ a.s. on the support of $\pi$. Since this holds for any $v \in \mathcal{F}$ and $f \in L^2(\mathcal{X}, \rho)$, this implies that $Av$ lies in the range of $L^{1/2}$ for all $v \in \mathcal{F}$. $\qquad\square$

Moreover, note that since $L_m = \mathfrak{A}C_m$ (See Proposition A.16), the eigenfunctions of $L_m$ are the lifted eigenvectors of $C_m$ from $\mathcal{F}$ to $L^2(\mathcal{X}, \rho)$. This equivalence entails that any candidate solution of OPT-2 has an equivalent candidate solution for OPT-4, and they would both have the same objective.

As already hinted, the solution of Kernel PCA with random features might not lie in the constraint set of rank $k$ projection operators over $L^2(\mathcal{X}, \rho)$. By the equivalence of the optimization problems OPT-1 and OPT-4, we violate the constraint in $\mathcal{H}$ as well. We, however, remarked that we counter this problem by showing a fast $O(1/\sqrt{n})$ speed of convergence to the constraint set in $L^2(\mathcal{X}, \rho)$. A natural question to ask is does the speed of convergence to the constraint set preserved too? We answer affirmatively as shown below.

We now use this equivalence to give a reduction from a candidate solution OPT-4 to OPT-2. Let $\tilde{P} = \sum_{i=1}^{k} \tilde{p}_i \otimes_\rho \tilde{p}_i$ be the output of some algorithm for Kernel PCA with random features, lifted through the approximation operator $\mathfrak{A}$. We have show in Theorem C.1 that $\mathfrak{A}P^k_{C_m} = P^k_{L_m}$ is a rank k projection over $L^2(\mathcal{X}, \rho)$. Let $P_{L_m} = \sum_{i=1}^{k} \tilde{q}_i \otimes \tilde{q}_i$. Since $\tilde{p}_i$ and $\tilde{q}_i$ lie in the range of $L^{1/2}$, $\forall i \in [k]$ (See Lemma B.5), there exists $p_i$'s and $q_i$'s such that $\tilde{p}_i = L^{1/2}p_i$ and $\tilde{q}_i = L^{1/2}q_i$, $i \in [k]$. Define $P := \sum_{i=1}^{k} I^*p_i \otimes_\mathcal{H} I^*p_i$, and $Q := \sum_{i=1}^{k} I^*q_i \otimes_\mathcal{H} I^*q_i$.

First we show that $Q$ is a projection operator in $HS(\mathcal{H})$.

$$\begin{aligned}
\langle I^*q_i, I^*q_j \rangle_\mathcal{H} &= \langle q_i, II^*q_j \rangle_\rho & \text{(Definition of adjoints)} \\
&= \langle q_i, Lq_j \rangle_\rho & \text{(Proposition A.9)} \\
&= \left\langle L^{1/2}q_i, L^{1/2}q_j \right\rangle_\rho & \text{(Definition of adjoints)} \\
&= \langle \tilde{q}_i, \tilde{q}_j \rangle_\rho = \delta_{ij}
\end{aligned}$$

Now, let us look at the rate of convergence $P$ to $\mathcal{P}^k_{HS(\mathcal{H})}$.

**Lemma B.6** (Equivalence of convergence to the constraint set).

$$d(\bar{P}, \mathcal{P}^k_{HS(\mathcal{H})}) \leq \left\| \tilde{P} - \mathfrak{A}C_m \right\|_{HS(\rho)}$$

*Proof.*

$$d(\bar{\mathrm{P}}, \mathcal{P}^k_{HS(\mathcal{H})}) \leq \|\mathrm{P} - \mathrm{Q}\|_{HS(\mathcal{H})}$$

$$= \left\| \sum_{i=1}^{k} \mathrm{I}^* \mathrm{p}_i \otimes_{\mathcal{H}} \mathrm{I}^* \mathrm{p}_i - \mathrm{I}^* \mathrm{q}_i \otimes_{\mathcal{H}} \mathrm{I}^* \mathrm{q}_i \right\|_{HS(\mathcal{H})}$$

$$= \left\| \mathrm{I}(\sum_{i=1}^{k} \mathrm{p}_i \otimes_{\rho} \mathrm{p}_i - \mathrm{q}_i \otimes_{\rho} \mathrm{q}_i)\mathrm{I}^* \right\|_{HS(\mathcal{H})} \qquad \text{(Proposition A.18)}$$

$$= \left\| \mathrm{L}(\sum_{i=1}^{k} \mathrm{p}_i \otimes_{\rho} \mathrm{p}_i - \mathrm{q}_i \otimes_{\rho} \mathrm{q}_i) \right\|_{HS(\rho)}$$

$$= \left\| \mathrm{L}^{1/2}(\sum_{i=1}^{k} \mathrm{p}_i \otimes_{\rho} \mathrm{p}_i - \mathrm{q}_i \otimes_{\rho} \mathrm{q}_i)\mathrm{L}^{1/2} \right\|_{HS(\rho)} \qquad \text{(Cyclic property)}$$

$$= \left\| \sum_{i=1}^{k} \mathrm{L}^{1/2}\mathrm{p}_i \otimes_{\rho} \mathrm{L}^{1/2}\mathrm{p}_i - \mathrm{L}^{1/2}\mathrm{q}_i \otimes_{\rho} \mathrm{L}^{1/2}\mathrm{q}_i \right\|_{HS(\rho)} \qquad \text{(Proposition A.18)}$$

$$= \left\| \sum_{i=1}^{k} \tilde{\mathrm{p}}_i \otimes_{\rho} \tilde{\mathrm{p}}_i - \tilde{\mathrm{q}}_i \otimes_{\rho} \tilde{\mathrm{q}}_i \right\|_{HS(\rho)}$$

$$= \left\| \tilde{\mathrm{P}} - \mathfrak{A}\mathrm{C}_m \right\|_{HS(\rho)}$$

$\square$

In Lemma C.5, we will bound $\left\| \tilde{\mathrm{P}} - \mathfrak{A}\mathrm{C}_m \right\|_{HS(\rho)}$ which implies the bound given in the main theorem 4.2.

We now combine the above relations into a definition to lift operators from $HS(\mathcal{F})$ to $HS(\mathcal{H})$ and then discuss that the operator is well-defined.

**Definition B.7** (Operator $\mathfrak{L}$). Let $\tilde{\mathrm{P}} \in HS(\mathcal{F})$ and $\mathfrak{A}\tilde{\mathrm{P}} = \sum_{i=1}^{k} \tilde{\mathrm{p}}_i \otimes_{\rho} \tilde{\mathrm{p}}_i$ be $\tilde{\mathrm{P}}$ lifted to $L^2(\mathcal{X}, \rho)$. Consider the equivalence relation $\mathrm{p}_i \sim \mathrm{p}_j$ if $\mathrm{L}^{1/2}\mathrm{p}_i = \mathrm{L}^{1/2}\mathrm{p}_j$. Let $[\mathrm{p}_i]$ be the equivalence class such that $\mathrm{L}^{1/2}\mathrm{p}_i = \tilde{\mathrm{p}}_i$. The operator $\mathfrak{L} : HS(\mathcal{F}) \to HS(\mathcal{H})$ is defined as,

$$\mathfrak{L}\widehat{P} = \sum_{i=1}^{k} \mathrm{I}^* \mathrm{p}_i \otimes_{\mathcal{H}} \mathrm{I}^* \mathrm{p}_i$$

Here $\mathrm{I}^*$ is the restriction of the operator $\mathrm{I}^*$ to the quotient space $L^2(\mathcal{X}, \rho)/\sim$.

We now discuss that the operator $\mathfrak{L}$ is indeed a well defined operator. We guarantee by Lemma B.5 that there is at least one element in $[\mathrm{p}_i]$ such that $\tilde{\mathrm{p}}_i = \mathrm{L}^{1/2}\mathrm{p}_i$. It remains to argue that all the elements in the equivalence class $[\mathrm{p}_i]$ are being mapped to the same element in $\mathcal{H}$ through $\mathrm{I}^*$. Let $\mathrm{p}_i$ and $\mathrm{p}_j$ be two elements of $[\mathrm{p}_i]$. Since $\mathrm{L}^{1/2}\mathrm{p}_i = \mathrm{L}^{1/2}\mathrm{p}_j$, therefore $\mathrm{L}^{1/2}(\mathrm{p}_i - \mathrm{p}_j) = 0$. This implies that $\mathrm{p}_j = \mathrm{p}_i + \mathrm{Ker}(\mathrm{L}^{1/2})$. Note that any $\mathrm{r}_i \in \mathrm{Ker}(\mathrm{L}^{1/2})$ will be mapped by $\mathrm{I}^*$ to 0, i.e. $\mathrm{I}^*\mathrm{r}_i = 0$. It follows from linearity of $\mathrm{I}^*$ that $\mathrm{I}^*\tilde{\mathrm{p}}_j = \mathrm{I}^*\mathrm{p}_i$. Thus this maps an equivalence class to a single element in $\mathcal{H}$.

## C  Proof of the main Theorem

From we have already established the problems in $HS(\rho)$ and $HS(\mathcal{H})$, we focus on error decomposition and bounding the corresponding error terms in $L^2(\mathcal{X}, \rho)$. Solving KPCA by using a kernel approximation, one needs to consider two different sources of error. First, the error coming from approximating the true kernel operator by random features. Second, the statistical error due to estimating the covariance operator using iid samples from the unknown distribution. Thus, we distinguish between and base our proof around these two sources of error, namely *approximation error* and *estimation error*. In particular we decompose our objective as:

$$\left\langle \mathfrak{I}\mathrm{P}_{\mathrm{C}}^k, \mathfrak{I}\mathrm{C} \right\rangle_{HS(\rho)} - \left\langle \mathfrak{A}\widehat{\mathrm{P}}, \mathfrak{I}\mathrm{C} \right\rangle_{HS(\rho)} = \underbrace{\left\langle \mathfrak{I}\mathrm{P}_{\mathrm{C}}^k, \mathfrak{I}\mathrm{C} \right\rangle_{HS(\rho)} - \left\langle \mathfrak{A}\mathrm{P}_{\mathrm{C}_m}^k, \mathfrak{I}\mathrm{C} \right\rangle_{HS(\rho)}}_{\epsilon_a:\ \text{Approximation Error}}$$

$$+ \underbrace{\left\langle \mathfrak{A}\mathrm{P}_{\mathrm{C}_m}^k, \mathfrak{I}\mathrm{C} \right\rangle_{HS(\rho)} - \left\langle \mathfrak{A}\widehat{\mathrm{P}}, \mathfrak{I}\mathrm{C} \right\rangle_{HS(\rho)}}_{\epsilon_e:\ \text{Estimation Error}}.$$

The first term in the decomposition is interpreted as approximation error because it essentially captures the error incurred by approximating the kernel function with random features. The second term in the decomposition is interpreted as estimation error as it is the error incurred in the original statistical estimation problem. In what follows, we give a bound on each of the error terms and provide a detailed analysis. Throughout this section, we use the following Lemma that shows the relation between different projection operators.

**Lemma C.1.** $\mathfrak{I}\mathrm{P}_{\mathrm{C}}^k$ *and* $\mathfrak{A}\mathrm{P}_{\mathrm{L}}^k$ *are rank $k$ projection operators in $L^2(\mathcal{X}, \rho)$. Furthermore, it holds that* $\mathfrak{I}\mathrm{P}_{\mathrm{C}}^k = \mathrm{P}_{\mathrm{L}}^k$ *and* $\mathfrak{A}\mathrm{P}_{\mathrm{C}_m}^k = \mathrm{P}_{\mathrm{L}_m}^k$.

*Proof of Lemma C.1.* We have

$$\mathfrak{I}\mathrm{P}_{\mathrm{C}}^k = \sum_{i=1}^k \frac{\mathrm{I}\bar{\phi}_i}{\sqrt{\bar{\lambda}_i}} \otimes_\rho \frac{\mathrm{I}\bar{\phi}_i}{\sqrt{\bar{\lambda}_i}} = \mathrm{P}_{\mathrm{L}}^k$$

where the second inequality follows from Lemma A.9. Similarly,

$$\mathfrak{A}\mathrm{P}_{\mathrm{C}_m}^k = \sum_{i=1}^k \frac{\mathrm{A}\phi_i}{\sqrt{\lambda_i}} \otimes_\rho \frac{\mathrm{A}\phi_i}{\sqrt{\lambda_i}} = \mathrm{P}_{\mathrm{L}_m}^k$$

where the second inequality follows from Lemma A.12. and $\mathfrak{A}\mathrm{P}_{\mathrm{C}}^k$ $\square$

### C.1  Approximation Error

The main idea behind controlling the approximation error is to use the local Rademacher complexity of the kernel class Massart [2000], Bartlett et al. [2002]. More precisely, we use the following result in [Blanchard et al., 2007], which allows us to get rates depending both on the number of features used and the decay of the spectrum of the operator $\mathrm{C}_2$.

**Theorem C.2** (Blanchard et al. [2007]). *Assume* $\|\zeta\|^2 \leq M$ *almost surely, and let* $(\lambda_i)$ *denote the ordered eigenvalues of* $\mathrm{C} := \mathrm{E}[\zeta\zeta^\top]$, *and further assume that* $(\lambda_i)$ *are distinct. Let* $B_k := \frac{\sqrt{\mathrm{E}[\langle \zeta, \zeta' \rangle^4]}}{\lambda_k - \lambda_{k+1}}$, *where $\zeta'$ is and iid copy of $\zeta$. Then for all $\delta$, with overwhelming probability of at least $1 - e^{-\delta}$ it holds that*

$$\langle P_{\widehat{\mathrm{C}}^\perp}^k, \mathrm{C} \rangle - \langle P_{\mathrm{C}^\perp}^k, \mathrm{C} \rangle \leq 24\kappa(B_k, k, n) + \frac{11\delta(M + B_k)}{n}$$

*where $\kappa$ is defined as follows:*

$$\kappa(B_k, k, n) = \inf_{h \geq 0} \left\{ \frac{B_k h}{n} + \sqrt{\frac{k}{n} \sum_{j > h} \lambda_i(\mathrm{C}')} \right\}$$

**Lemma C.3** (Approximation Error). *With probability at least $1 - \frac{\delta}{2}$, we have*

$$\langle \mathrm{P}_{\mathrm{L}_m^\perp}^k, \mathrm{L} \rangle - \langle \mathrm{P}_{\mathrm{L}^\perp}^k, \mathrm{L} \rangle \leq 24\kappa(B_k, k, m) + \frac{11 \log(\delta/2)\, \tau^2 + 7B_k}{m}$$

*Proof.* We first note that $\mathfrak{A}P_{C_m}^k = \sum_{i=1}^k \frac{A\phi_i}{\sqrt{\lambda_i}} \otimes \frac{A\phi_i}{\sqrt{\lambda_i}}$ (see definition of the approximation operator in A.10). The following holds for the approximation error:

$$\epsilon_a = \left\langle \mathfrak{I}P_C^k, \mathfrak{I}C \right\rangle_{HS(\rho)} - \left\langle \mathfrak{A}P_{C_m}^k, \mathfrak{I}C \right\rangle_{HS(\rho)} \tag{5}$$

$$= \left\langle \sum_{i=1}^k \frac{I\bar{\phi}_i}{\sqrt{\bar{\lambda}_i}} \otimes_\rho \frac{I\bar{\phi}_i}{\sqrt{\bar{\lambda}_i}}, \sum_{i\in I\subset\mathbb{R}} \bar{\lambda}_i \left( \frac{I\bar{\phi}_i}{\sqrt{\bar{\lambda}_i}} \otimes_\rho \frac{I\bar{\phi}_i}{\sqrt{\bar{\lambda}_i}} \right) \right\rangle_{HS(\rho)} \quad \text{(definition A.7)}$$

$$- \left\langle \sum_{i=1}^k \frac{A\phi_i}{\sqrt{\lambda_i}} \otimes_\rho \frac{A\phi_i}{\sqrt{\lambda_i}}, \sum_{i\in I\subset\mathbb{R}} \bar{\lambda}_i \left( \frac{I\bar{\phi}_i}{\sqrt{\bar{\lambda}_i}} \otimes_\rho \frac{I\bar{\phi}_i}{\sqrt{\bar{\lambda}_i}} \right) \right\rangle_{HS(\rho)} \quad \text{(definition A.7, A.10)}$$

$$= \langle P_L^k, L\rangle_{HS(\rho)} - \langle P_{L_m}^k, L\rangle_{HS(\rho)} \quad \text{(Lemma A.15 and Lemma C.1)}$$

$$= \langle P_{L_m^\perp}^k, L\rangle_{HS(\rho)} - \langle P_{L^\perp}^k, L\rangle_{HS(\rho)} \quad \text{(properties of the orthogonal subspace)}$$

We have already showed that L and $L_m$ in the right hand side of the equation (5) are true and empirical covariance operators respectively (see Lemma A.17). As required by Theorem C.2, we need to show that norm of the random variables $z_\omega$ are bounded. We have

$$\|z_\omega\|^2 = \langle z_\omega, z_\omega\rangle_\rho$$
$$= \int_{\mathcal{X}} z_\omega(\mathrm{x},t)^2 d\rho(\mathrm{x},t)$$
$$\leq \tau^2$$

where the last inequality follows Assumption 3.1.

Invoking Theorem C.2, we have with probability at least $1-\delta$,

$$\langle P_{L_m^\perp}^k, L\rangle - \langle P_{L^\perp}^k, L\rangle \leq 24\kappa(B_k, k, m) + \frac{11\log(\delta)\tau^2 + 7B_k}{m}$$

where $\kappa(B_k, k, m) = \inf_{h\geq 0}\left\{ \frac{B_k h}{m} + \sqrt{\frac{k\sum_{j>h}\lambda_j(C_2')}{m}} \right\}$. $\qquad\square$

**Lemma C.4** (Approximation Error - Good decay). *When the spectrum of operator $C_2'$ has an exponential decay, i.e. $\lambda_j(C_2') = \alpha^j$ for some $\alpha < 1$, then with probability at least $1-\delta$, we have*

$$\langle P_{L_m^\perp}^k, L\rangle - \langle P_{L^\perp}^k, L\rangle \leq \frac{24B_k\log(m)}{\log(1/\alpha)m} + \frac{k + (1-\alpha)(11\log(\delta)\tau^2 + 7B_k)}{(1-\alpha)m}$$

*Proof.* When $\lambda_i(C_2')$ have an exponential decay, i.e $\lambda_j(C_2') = \alpha^j$ for some $\alpha < 1$, we have

$$\sum_{j>h}\lambda_j(C_2') = \frac{\alpha^{h+1}}{1-\alpha}$$

Set $h = \lceil -\log_\alpha(m)\rceil - 1$, we get

$$\sum_{j>h}\lambda_j(C_2') \leq \frac{1}{(1-\alpha)m}$$

Now,

$$\kappa(B_k, k, m) = \inf_{h\geq 0}\left\{ \frac{B_k h}{m} + \sqrt{\frac{k\sum_{j>h}\lambda_j(C_2')}{m}} \right\}$$

$$\leq \frac{-B_k\log_\alpha m}{m} + \frac{k}{(1-\alpha)m}$$

$$= \frac{B_k\log(m)}{\log(1/\alpha)m} + \frac{k}{(1-\alpha)m}$$

where the last equality follows from the identity $\log_b a = \dfrac{\log_d(a)}{\log_d(b)}$

So essentially, $\kappa(B_k, k, m) = O\left(\frac{\log(m)}{m}\right)$. Therefore, we get

$$
\begin{aligned}
\epsilon_a = \left\langle \mathfrak{J}P_{\mathrm{C}}^k, \mathfrak{J}\mathrm{C} \right\rangle_{HS(\rho)} - \left\langle \mathfrak{A}P_{\mathrm{C}_m}^k, \mathfrak{J}\mathrm{C} \right\rangle_{HS(\rho)} &\leq \frac{24 B_k \log(m)}{\log(1/\alpha)\, m} + \frac{k}{(1-\alpha)m} + \frac{11 \log(\delta)\, \tau^2 + 7 B_k}{m} \\
&= \frac{24 B_k \log(m)}{\log(1/\alpha)\, m} + \frac{k + (1-\alpha)(11 \log(\delta)\, \tau^2 + 7 B_k)}{(1-\alpha)m}
\end{aligned}
$$

which completes the proof. $\qquad\qquad\qquad\qquad\qquad\qquad\qquad\qquad\qquad\qquad\qquad\qquad\quad$ $\square$

## C.2 Estimation Error

We first remind the reader that $\mathfrak{A}P_{\mathrm{C}_m}^k$ is a projection operator in $L^2(\mathcal{X}, \rho)$ (See Lemma C.1). However, the problem we face is that $\mathfrak{A}\widehat{P}$ might not be a projection operator in $L^2(\mathcal{X}, \rho)$. This is because the lifting is accomplished by lifting a particular set of eigenvectors of $\widehat{\mathrm{P}}_{\mathcal{A}}$ through A, and we remark that A doesn't necessarily preserve norms and angles between elements. To get around this predicament, we show that lifted operator converges to a projection operator, i.e the lifted set of eigenvectors go to an orthogonal set of functions in $L^2(\mathcal{X}, \rho)$. Moreover, from Lemma B.6, we have that this convergence in $HS(\rho)$ is equivalent to convergence in $HS(\mathcal{H})$.

**Lemma C.5.** *When the number of samples* $n \geq \frac{2\lambda_1^2 q_{\mathcal{A}}(1/\delta, \log(m), \log(n))^2}{\lambda_k^2(\sqrt{2}-1)}$, *with probability at least* $1 - \frac{\delta}{2}$, *we have*

$$
\begin{aligned}
d(\mathfrak{A}\mathrm{C}_m, \mathcal{P}_{HS(\rho)}^k) \leq \left\| \mathfrak{A}P_{\mathrm{C}_m}^k - \mathfrak{A}\widehat{P}_{\mathcal{A}} \right\|_{HS(\rho)} &\leq \left\| \mathfrak{A}P_{\mathrm{C}_m}^k - \mathfrak{A}\widehat{P}_{\mathcal{A}} \right\|_{\mathcal{L}^1(\rho)} \\
&\leq \frac{\lambda_1}{(\sqrt{2}-1)} \sqrt{\sum_{i=1}^k \left( \frac{2\lambda_i + 4\lambda_1}{\lambda_i^2} \right)^2 \frac{q_{\mathcal{A}}(1/\delta, \log(m), \log(n))}{n}}
\end{aligned}
$$

*Proof of Lemma C.5.* Since $\mathfrak{A}\mathrm{C}_m$ is a rank $k$ projection operator in $HS(\rho)$ (from Lemma C.1), the first inequality follows trivially. The second inequality is just from the property of norms that schatten norms $\|\mathrm{D}\|_{\mathcal{L}^p(\rho)}$ decreases with increasing $p$. We focus on proving the third inequality below.

Let $\widehat{\mathrm{P}}_{\mathcal{A}} = \tilde{\Phi}\tilde{\Phi}^\top$ be an eigendecomposition of the output $\widehat{\mathrm{P}}_{\mathcal{A}}$. Let

$$
\mathrm{R}^* = \underset{\mathrm{R}^\top \mathrm{R} = \mathrm{RR}^\top = \mathrm{I}}{\arg\min} \left\| \tilde{\Phi}\mathrm{R} - \Phi_k \right\|_F^2
$$

where $\Phi_k$ is the matrix corresponding top top $k$ eigenvectors of $\mathrm{C}_m$. Define $\widehat{\Phi} := \tilde{\Phi}\mathrm{R}^*$. This means that we rotate the eigenvectors of our output to a basis such that it is closest to the truth (in element-wise metric sense). An important point on why we can do this is that this rotation (or any other rotation for that matter) doesn't change the output, i.e. $\widehat{\Phi}\widehat{\Phi}^\top = \tilde{\Phi}\mathrm{R}^*\mathrm{R}^{*\top}\tilde{\Phi}^\top = \tilde{\Phi}\tilde{\Phi}^\top = \widehat{\mathrm{P}}_{\mathcal{A}}$. We now lifting the output by lifting this rotated set of eigenvectors. We have, $\mathfrak{A}\widehat{P} = \sum_{i=1}^k \frac{A\widehat{\phi}_i}{\sqrt{\widehat{\lambda}_i}} \otimes_\rho \frac{A\widehat{\phi}_i}{\sqrt{\widehat{\lambda}_i}}$, where $\widehat{\lambda}_i := \left\langle \widehat{\phi}_i, \mathrm{C}_m\widehat{\phi}_i \right\rangle_{\mathcal{F}}$.

$$\left\| \mathfrak{A} P_{C_m}^k - \mathfrak{A} \widehat{P}_{\mathcal{A}} \right\|_{\mathcal{L}^1(\rho)} = \left\| \sum_{i=1}^{k} \frac{\mathrm{A}\phi_i}{\sqrt{\lambda_i}} \otimes \frac{\mathrm{A}\phi_i}{\sqrt{\lambda_i}} - \sum_{i=1}^{k} \frac{\mathrm{A}\widehat{\phi}_i}{\sqrt{\widehat{\lambda}_i}} \otimes \frac{\mathrm{A}\widehat{\phi}_i}{\sqrt{\widehat{\lambda}_i}} \right\|_{\mathcal{L}^1(\rho)}$$

$$= \left\| \mathrm{A} \sum_{i=1}^{k} \left( \frac{\phi_i}{\sqrt{\lambda_i}} \otimes \frac{\phi_i}{\sqrt{\lambda_i}} - \frac{\widehat{\phi}_i}{\sqrt{\widehat{\lambda}_i}} \otimes \frac{\widehat{\phi}_i}{\sqrt{\widehat{\lambda}_i}} \right) \mathrm{A}^* \right\|_{\mathcal{L}^1(\rho)}$$

$$\leq \|\mathrm{A}\| \left\| \sum_{i=1}^{k} \left( \frac{1}{\lambda_i} \phi_i \otimes \phi_i - \frac{1}{\widehat{\lambda}_i} \widehat{\phi}_i \otimes \widehat{\phi}_i \right) \mathrm{A}^* \right\|_{\mathcal{L}^1(\mathcal{F},\rho)}$$

$$\leq \|\mathrm{A}\| \, \|\mathrm{A}^*\| \left\| \sum_{i=1}^{k} \left( \frac{1}{\lambda_i} \phi_i \otimes \phi_i - \frac{1}{\widehat{\lambda}_i} \widehat{\phi}_i \otimes \widehat{\phi}_i \right) \right\|_{\mathcal{L}^1(\mathcal{F})}$$

$$\leq \lambda_1 \left\| \sum_{i=1}^{k} \left( \frac{1}{\lambda_i} \phi_i \otimes \phi_i - \frac{1}{\widehat{\lambda}_i} \widehat{\phi}_i \otimes \widehat{\phi}_i \right) \right\|_{\mathcal{L}^1(\mathcal{F})}$$

Where third and fourth inequalities follows from the fact that for trace-class operators $\|AB\|_{\mathcal{L}^1} \leq \|A\|_2 \|B\|_{\mathcal{L}^1}$. See [Reed and Simon, 1972, Exercise 28, Page 218].

Adding and subtracting $\frac{1}{\lambda_i} \widehat{\phi}_i \otimes \widehat{\phi}_i$ inside the summation to get

$$\leq \lambda_1 \left\| \sum_{i=1}^{k} \frac{1}{\lambda_i} \phi_i \otimes \phi_i - \frac{1}{\lambda_i} \widehat{\phi}_i \otimes \widehat{\phi}_i + \frac{1}{\lambda_i} \widehat{\phi}_i \otimes \widehat{\phi}_i - \frac{1}{\widehat{\lambda}_i} \widehat{\phi}_i \otimes \widehat{\phi}_i \right\|_{\mathcal{L}^1(\mathcal{F})}$$

$$\leq \lambda_1 \left\| \sum_{i=1}^{k} \left( \frac{1}{\lambda_i} \left( \phi_i \otimes \phi_i - \widehat{\phi}_i \otimes \widehat{\phi}_i \right) + \left( \frac{1}{\lambda_i} - \frac{1}{\widehat{\lambda}_i} \right) \widehat{\phi}_i \otimes \widehat{\phi}_i \right) \right\|_{\mathcal{L}^1(\mathcal{F})}$$

$$\leq \lambda_1 \sum_{i=1}^{k} \frac{1}{\lambda_i} \left\| \phi_i \otimes \phi_i - \widehat{\phi}_i \otimes \widehat{\phi}_i \right\|_{\mathcal{L}^1(\mathcal{F})} + \left| \frac{1}{\lambda_i} - \frac{1}{\widehat{\lambda}_i} \right|$$

$$\leq \lambda_1 \sum_{i=1}^{k} \frac{1}{\lambda_i} \left\| \phi_i \otimes \phi_i - \widehat{\phi}_i \otimes \widehat{\phi}_i \right\|_{\mathcal{L}^1(\mathcal{F})} + \left| \frac{\lambda_i - \widehat{\lambda}_i}{\lambda_i \widehat{\lambda}_i} \right|$$

$$\leq \lambda_1 \sum_{i=1}^{k} \frac{2}{\lambda_i} \left\| \phi_i - \widehat{\phi}_i \right\|_2 + \frac{4\lambda_1}{\lambda_i^2} \left\| \phi_i - \widehat{\phi}_i \right\|_2$$

$$\leq \lambda_1 \sum_{i=1}^{k} \left( \frac{2\lambda_i + 4\lambda_1}{\lambda_i^2} \right) \left\| \phi_i - \widehat{\phi}_i \right\|_2$$

$$\leq \lambda_1 \sqrt{ \sum_{i=1}^{k} \left( \frac{2\lambda_i + 4\lambda_1}{\lambda_i^2} \right)^2 } \left\| \Phi_k - \widehat{\Phi} \right\|_{\mathcal{F}}$$

$$\leq \frac{\lambda_1}{2(\sqrt{2}-1)} \sqrt{ \sum_{i=1}^{k} \left( \frac{2\lambda_i + 4\lambda_1}{\lambda_i^2} \right)^2 } \left\| \mathrm{P}_{C_m}^k - \widehat{\mathrm{P}} \right\|_F^2$$

$$\leq \frac{\lambda_1}{(\sqrt{2}-1)} \sqrt{ \sum_{i=1}^{k} \left( \frac{2\lambda_i + 4\lambda_1}{\lambda_i^2} \right)^2 \frac{q_{\mathcal{A}}(1/\delta, \log(m), \log(n))}{n} }$$

The second to last inequality follows from Lemma C.6 and Lemma C.7. $\qquad\square$

**Lemma C.6.** $\|\phi_i \otimes \phi_i - \widehat{\phi}_i \otimes \widehat{\phi}_i\|_{\mathcal{L}^1(\mathcal{F})} \leq 2\|\phi_i - \widehat{\phi}_i\|_2 \; \forall \, i \in [k]$

*Proof.*

$$\|\phi_i \otimes \phi_i - \widehat{\phi}_i \otimes \widehat{\phi}_i\|_{\mathcal{L}^1(\mathcal{F})} = \|\phi_i \otimes \phi_i - \widehat{\phi}_i \otimes \phi_i + \widehat{\phi}_i \otimes \phi_i - \widehat{\phi}_i \otimes \widehat{\phi}_i\|_{\mathcal{L}^1(\mathcal{F})}$$
$$\leq \|(\phi_i - \widehat{\phi}_i) \otimes \phi_i\|_{\mathcal{L}^1(\mathcal{F})} + \|\widehat{\phi}_i \otimes (\phi_i - \widehat{\phi}_i)\|_{\mathcal{L}^1(\mathcal{F})}$$
$$= 2\|\phi_i - \widehat{\phi}_i\|_2$$

$\square$

**Lemma C.7.** *When the number of samples* $n \geq \frac{2q_{\mathcal{A}}(1/\delta, \log(m), \log(n))^2 \lambda_1^2}{\lambda_i^2(\sqrt{2}-1)}$, *with probability at least* $1 - \delta$, $\forall \, i \in [k]$ *we have,*

$$\left| \frac{\lambda_i - \widehat{\lambda}_i}{\lambda_i \widehat{\lambda}_i} \right| \leq \frac{4\lambda_1}{\lambda_i^2} \left\| \phi_i - \widehat{\phi}_i \right\|_2$$

*where* $C_{\mathcal{A}}$ *is a constant specific to the algorithm* $\mathcal{A}$.

The numerator is bounded as follows

*Proof.*

$$|\lambda_i - \widehat{\lambda}_i| = |\phi_i^\top C_m \phi_i - \widehat{\phi}_i^\top C_m \widehat{\phi}_i|$$
$$= |\phi_i^\top C_m \phi_i - \phi_i^\top C_m \widehat{\phi}_i + \phi_i^\top C_m \widehat{\phi}_i - \widehat{\phi}_i^\top C_m \widehat{\phi}_i|$$
$$= |\phi_i^\top C_m (\phi_i - \widehat{\phi}_i) + (\phi_i - \widehat{\phi}_i)^\top C_m \widehat{\phi}_i|$$
$$\leq \|C_m \phi_i\|_2 \|\phi_i - \widehat{\phi}_i\|_2 + \|\phi_i - \widehat{\phi}_i\|_2 \|C_m \widehat{\phi}_i\|_2$$
$$= (\lambda_i + \widehat{\lambda}_i)\|\phi_i - \widehat{\phi}_i\|_2$$
$$\leq (\lambda_i + \lambda_1)\|\phi_i - \widehat{\phi}_i\|_2$$
$$\leq 2\lambda_1 \|\phi_i - \widehat{\phi}_i\|$$

where the second inequality holds since $\widehat{\lambda}_i < \lambda_1$ by definition of $\widehat{\lambda}_i$, and the last inequality follows because $\widehat{\lambda}_i \leq \lambda_1$.

The denominator is lower bounded similarly as

$$\lambda_i \widehat{\lambda}_i \geq \lambda_i(\lambda_i - 2\lambda_1 \|\phi_i - \widehat{\phi}_i\|)$$
$$\geq \frac{\lambda_i^2}{2}$$

where the first inequality follows from the bound on the numerator and the last inequality follows when $2\lambda_1 \left\| \phi_i - \widehat{\phi}_i \right\| \leq \frac{\lambda_i}{2}$. From Lemma C.10, we know that $\left\| \phi_i - \widehat{\phi}_i \right\|_2^2 \leq \frac{1}{2(\sqrt{2}-1)} \left( \frac{q_{\mathcal{A}}(1/\delta, \log(m), \log(n))}{n} \right)$ with probability at least $1 - \delta$. Combining, we get, with probability at least $1 - \delta$,

$$\left\| \phi_i - \widehat{\phi}_i \right\|_2 \leq \sqrt{\frac{1}{2(\sqrt{2}-1)} \left( \frac{q_{\mathcal{A}}(1/\delta, \log(m), \log(n))}{n} \right)} \leq \frac{\lambda_i}{4\lambda_1}$$

The above holds when the number of samples $n \geq \frac{2\lambda_1^2 q_{\mathcal{A}}(1/\delta, \log(m), \log(n))^2}{\lambda_i^2(\sqrt{2}-1)}$. Combining, we get

$$\left| \frac{\lambda_i - \widehat{\lambda}_i}{\lambda_i \widehat{\lambda}_i} \right| \leq \frac{4\lambda_1}{\lambda_i^2} \left\| \phi_i - \widehat{\phi}_i \right\|_2$$

with probability at least $1 - \delta$ and when $n \geq \frac{2\lambda_1^2 q_{\mathcal{A}}(1/\delta, \log(m), \log(n))^2}{\lambda_i^2(\sqrt{2}-1)}$. $\square$

Note that in particular since Oja's algorithm has a warm-up phase, the lower bound on the denominator

**Lemma C.8.** *For rank $k$ orthogonal matrices $U \in \mathbb{R}^{m \times k}$ and $V \in \mathbb{R}^{m \times k}$, i.e. $U^\top U = V^\top V = I_k$, the following holds,*

$$\left\| U - V\widehat{R} \right\|_F^2 \leq \frac{1}{2(\sqrt{2}-1)} \left\| UU^\top - VV^\top \right\|_F^2,$$

*where*

$$\widehat{R} = \underset{R^\top R = RR^\top = I_k}{\arg\min} \|U - VR\|_F^2$$

*Proof.* Proof in [Ge et al., 2017, Lemma 6]. □

Since $\Phi_k$ and $\widehat{\Phi}$ are rank $k$ orthogonal matrices, from Lemma C.8, we have

$$\left\| \Phi_k - \widehat{\Phi} \right\|_F^2 \leq \frac{1}{2(\sqrt{2}-1)} \left\| P_{C_m}^k - \widehat{P} \right\|_F^2$$

**Lemma C.9.** *For any efficient subspace learner $\mathcal{A}$, we have $\left\| P_{C_m}^k - \widehat{P} \right\|_F^2 \leq \frac{2q_{\mathcal{A}}(1/\delta, \log(m), \log(n))}{n}$ with probability at least $1 - \frac{\delta}{2}$.*

*Proof.*

$$\left\| P_{C_m}^k - \widehat{P}_{\mathcal{A}} \right\|_F^2 = \left\| P_{C_m}^k \right\|_F^2 + \left\| \widehat{P} \right\|_F^2 - 2\left\langle \widehat{P}, P_{C_m}^k \right\rangle$$

$$= 2\left( k - \left\langle \widehat{P}, P_{C_m}^k \right\rangle \right)$$

$$= 2\left( \left\langle I - P_{C_m}^k, \widehat{P} \right\rangle \right)$$

$$= 2\left\| \left(\Phi_k^\perp\right)^\top \widehat{\Phi} \right\|_F^2$$

$$\leq \frac{2q_{\mathcal{A}}(1/\delta, \log(m), \log(n))}{n}$$

where the last inequality follows from the definition of efficient subspace learner. □

**Lemma C.10.** *With probability at least $1 - \delta$,*

$$\left\| \phi_i - \widehat{\phi}_i \right\|_2 \leq \frac{1}{2(\sqrt{2}-1)} \left( \frac{q_{\mathcal{A}}(1/\delta, \log(m), \log(n))}{n} \right)$$

*where $q_{\mathcal{A}}(1/\delta, \log(m), \log(n))$ is specific to the algorithm $\mathcal{A}$.*

*Proof.*

$$\left\| \phi_i - \widehat{\phi}_i \right\|_2^2 \leq \sum_{i=1}^k \left\| \phi_i - \widehat{\phi}_i \right\|_2^2$$

$$= \left\| \Phi_k - \widehat{\Phi} \right\|_F^2$$

$$\leq \frac{1}{2(\sqrt{2}-1)} \left\| P_{C_m}^k - \widehat{P}_{\mathcal{A}} \right\|_F^2$$

$$\leq \frac{1}{2(\sqrt{2}-1)} \left( \frac{q_{\mathcal{A}}(1/\delta, \log(m), \log(n))}{n} \right)$$

□

where the second inequality holds from Lemma C.8 and the definition of $\widehat{P}$, and the last inequality holds from Lemma C.9

**Lemma C.11** (Estimation Error). *When the number of samples* $n \geq \dfrac{2\lambda_1^2 q_{\mathcal{A}}(1/\delta, \log{(m)}, \log{(n)})^2}{\lambda_k^2(\sqrt{2}-1)}$,
*then with probability at least* $1 - \delta$, *we have*

$$\epsilon_e \leq \frac{\lambda_1^2}{(\sqrt{2}-1)} \sqrt{\sum_{i=1}^{k}\left(\frac{2\lambda_i + 4\lambda_1}{\lambda_i^2}\right)^2 \frac{q_{\mathcal{A}}(1/\delta, \log{(m)}, \log{(n)})^2}{n}}$$

*Proof.*

$$
\begin{aligned}
\epsilon_e &= \langle \mathfrak{A}P_{C_m}^k, \mathfrak{I}C \rangle_{HS(\rho)} - \langle \mathfrak{A}\widehat{P}_{\mathcal{A}}, \mathfrak{I}C \rangle_{HS(\rho)} \\
&= \left\langle \mathfrak{A}P_{C_m}^k - \mathfrak{A}\widehat{P}_{\mathcal{A}}, \mathfrak{I}C \right\rangle_{HS(\rho)} \\
&\leq \left\| \mathfrak{A}P_{C_m}^k - \mathfrak{A}\widehat{P}_{\mathcal{A}} \right\|_{\mathcal{L}^1(\rho)} \|\mathfrak{I}C\|_2 \\
&\leq \lambda_1 \left\| \mathfrak{A}P_{C_m}^k - \mathfrak{A}\widehat{P}_{\mathcal{A}} \right\|_{\mathcal{L}^1(\rho)} \\
&\leq \frac{\lambda_1^2}{(\sqrt{2}-1)} \sqrt{\sum_{i=1}^{k}\left(\frac{2\lambda_i + 4\lambda_1}{\lambda_i^2}\right)^2 \frac{q_{\mathcal{A}}(1/\delta, \log{(m)}, \log{(n)})}{n}}
\end{aligned}
$$

The last inequality follows from Lemma C.5. $\qquad\square$

We now invoke the approximation and the estimation error bounds i.e. Lemma C.3 and Lemma C.11 with failure probabilities $\delta/2$ each. We then apply a union bound over them and get that with probability at least $1 - \delta$,

$$\langle \mathfrak{I}P_C^k, \mathfrak{I}C \rangle_\rho - \langle \mathfrak{A}\widehat{P}_{\mathcal{A}}, \mathfrak{I}C \rangle_\rho \leq \frac{cB_k}{\sqrt{n}} + \frac{c'(k + \log{(\delta/2)} + 7B_k)}{\sqrt{n}\log{(n)}} + \sqrt{\frac{q_{\mathcal{A}}(2/\delta, \log{(m)}, \log{(n)})}{n}},$$

This concludes the proof of the main theorem.

Also note that since $d(\mathfrak{A}\widehat{P}_{\mathcal{A}}, \mathcal{P}_{HS(\rho)})$ decays as $O(1/\sqrt{n})$, we can bound the suboptimality of $\mathfrak{A}\widehat{P}_{\mathcal{A}}$ projected onto the set of projection operators $\mathcal{P}_{HS(\rho)}^k$. It is now easy to give a bound on the objective with respect to the projection $\tilde{P}_{\mathcal{A}} \in \mathcal{P}_{HS(\rho)}$ of $\mathfrak{A}\widehat{P}_{\mathcal{A}}$ onto the set of projection operators:

**Corollary C.12.** *Let* $\tilde{P}_{\mathcal{A}}$ *be the projection of* $\mathfrak{A}\widehat{P}_{\mathcal{A}}$ *onto the set* $\mathcal{P}_{HS(\rho)}$. *Under the same conditions as in theorem 4.2, we have*

$$\langle \mathfrak{I}P_C^k, \mathfrak{I}C \rangle_\rho - \langle \tilde{P}_{\mathcal{A}}, \mathfrak{I}C \rangle_\rho \leq 24\kappa(B_k, k, m) + \frac{\log{(\delta/2)} + 7B_k}{m} + 2\sqrt{\frac{q_{\mathcal{A}}(2/\delta, \log{(m)}, \log{(n)})}{n}}$$

*Proof.*

$$
\begin{aligned}
\langle \mathfrak{I}P_C^k, \mathfrak{I}C \rangle_\rho - \langle \tilde{P}_{\mathcal{A}}, \mathfrak{I}C \rangle_\rho &= \langle \mathfrak{I}P_C^k, \mathfrak{I}C \rangle_\rho - \langle \mathfrak{A}\widehat{P}_{\mathcal{A}}, \mathfrak{I}C \rangle_\rho + \langle \mathfrak{A}\widehat{P}_{\mathcal{A}} - \tilde{P}_{\mathcal{A}}, \mathfrak{I}C \rangle \\
&\leq 24\kappa(B_k, k, m) + \frac{\log{(\delta/2)} + 7B_k}{m} + \sqrt{\frac{q_{\mathcal{A}}(2/\delta, \log{(m)}, \log{(n)})}{n}} \\
&\quad + d\left(\mathfrak{A}\widehat{P}_{\mathcal{A}}, \mathcal{P}_{HS(\rho)}\right) \|\mathfrak{I}C\|_{HS(\rho)} \\
&\leq 24\kappa(B_k, k, m) + \frac{\log{(\delta/2)} + 7B_k}{m} + 2\sqrt{\frac{q_{\mathcal{A}}(2/\delta, \log{(m)}, \log{(n)})}{n}},
\end{aligned}
$$

where the second to last inequality follows from Cauchy-Schwartz in $HS(\rho)$. $\qquad\square$

We now give the proof of Corollary 4.3.

*Proof of Corollary 4.3.*

$$\langle \mathfrak{I}\mathrm{P}_{\mathrm{C}}^k, \mathfrak{I}\mathrm{C}\rangle_\rho - \langle \mathfrak{A}\widehat{\mathrm{P}}_{\mathcal{A}}, \mathfrak{I}\mathrm{C}\rangle_\rho = \langle \mathfrak{I}\mathrm{P}_{\mathrm{C}}^k, \mathfrak{I}\mathrm{C}\rangle_{HS(\rho)} - \langle \mathfrak{A}\mathrm{P}_{\mathrm{C}_m}^k, \mathfrak{I}\mathrm{C}\rangle_{HS(\rho)}$$

$$+ \langle \mathfrak{A}\mathrm{P}_{\mathrm{C}_m}^k, \mathfrak{I}\mathrm{C}\rangle_{HS(\rho)} - \langle \mathfrak{A}\widehat{\mathrm{P}}, \mathfrak{I}\mathrm{C}\rangle_{HS(\rho)}$$

$$\leq \frac{24 B_k \log(m)}{\log(1/\alpha)\, m} + \frac{k + (1-\alpha)(11 \log(\delta/2)\, M + 7 B_k)}{(1-\alpha)m}$$

$$+ \lambda_1 \sqrt{\frac{q_{\mathcal{A}}(2/\delta, \log(m), \log(n))}{n}},$$

with probability at least $1 - \delta$. The last inequality follows from Lemma C.4 and Lemma C.11 with a union bound over them. $\qquad\square$

# D Examples of ESL

In this section, we instantiate our framework with two popular learning algorithms, Empirical Risk Minimization (ERM) and Oja's Algorithm, and show that they satisfy the requirements of ESL.

## D.1 Empirical Risk Minimizer

A natural candidate for an efficient subspace learner is the Empirical Risk Minimizer, which we call as $\mathcal{A}_{ERM}$. We first show that $\mathcal{A}_{ERM}$ satisfies the sufficient condition of Definition 4.1 and then show that $\mathcal{A}_{ERM}$ is an efficient subspace learner. We then discuss its computational aspects. Let $\{x_i\}_{i=1}^n$ be $n$ data samples and $\{z(x)\}_{i=1}^n$ be the corresponding representations in $\mathcal{F}$. The empirical covariance matrix in $\mathcal{F}$ is defined as

$$\widehat{C}_m = \frac{1}{n} \sum_{i=1}^n z(x_i)z(x_i)^\top$$

The algorithm $\mathcal{A}_{ERM}$ computes the top $k$ eigenvectors of $\widehat{C}_m$, and returns a rank $k$ orthogonal matrix say $\widehat{\Phi}$. Let the corresponding projection matrix be $\widehat{P}_{ERM}$. We first state the bound on covariance matrices $C_m$ and $\widehat{C}_m$.

**Lemma D.1** (Covariance Estimation). *With probability at least $1 - \delta$,*

$$\left\|\widehat{C}_m - C_m\right\|_2 \leq \frac{\kappa}{3n} \log\left(\frac{\delta}{2m}\right) + \sqrt{\frac{\kappa}{3n} \log\left(\frac{\delta}{2m}\right)^2 + \log\left(\frac{\delta}{2m}\right) \frac{\kappa \lambda_1}{n}}$$

*Proof.*

$$\left\|\widehat{C}_m - C_m\right\|_2 = \left\|\frac{1}{n} \sum_{i=1}^n z(x_i)z(x_i)^\top - C_m\right\|_2$$

$$= \left\|\frac{1}{n} \sum_{i=1}^n \left(z(x_i)z(x_i)^\top - C_m\right)\right\|_2$$

$$= \left\|\sum_{i=1}^n \Xi_i\right\|_2$$

where $\Xi_i = \frac{1}{n}\left(z(x_i)z(x_i)^\top - C_m\right)$. $\Xi_i$'s are 0 mean random matrices, i.e. $\mathbb{E}\left[\Xi_i\right] = 0 \,\forall\, i \in [n]$. Note that $\|z(x)\|_2^2 = \int_{\mathcal{X}} z_\omega(x)^2 d\rho(x) \leq \tau^2$, since $z_\omega(x) \leq \tau \,\forall\, \omega \in \Omega, x \in \mathcal{X}$ by Assumption 3.1. We have,

$$\|\Xi_i\|_2 \leq \frac{1}{n}\left(\left\|z(x_i)z(x_i)^\top\right\|_2 + \|C_m\|_2\right)$$

$$= \frac{1}{n}\left(\mathrm{Tr}\left(z(x_i)z(x_i)^\top\right) + \left\|\mathbb{E}_x\left[z(x)z(x)^\top\right]\right\|_2\right)$$

$$\leq \frac{1}{n}\left(\|z(x_i)\|^2 + \mathbb{E}_x\left[\left\|z(x)z(x)^\top\right\|\right]_2\right)$$

$$\leq \frac{1}{n}\left(\|z(x_i)\|^2 + \mathbb{E}_x\left[\|z(x)\|_2^2\right]\right)$$

$$\leq \frac{2\tau^2}{n}$$

so that $L(\Xi) := \max_i\{\|\Xi_i\|_2\} \leq \frac{2\tau^2}{n}$. where in the second inequality, we apply Jensen's inequality. Define $v(\Xi) := \left\|\sum_{i=1}^n \mathbb{E}\left[\Xi_i\Xi_i^\top\right]\right\|_2$. We have,

$$
\begin{aligned}
v(\Xi) &= \left\|\sum_{i=1}^n \mathbb{E}\left[\Xi_i\Xi_i^\top\right]\right\|_2 \\
&= \left\|\sum_{i=1}^n \mathbb{E}\left[\frac{1}{n^2}\left(\mathrm{z}(\mathrm{x}_i)\mathrm{z}(\mathrm{x}_i)^\top - \mathrm{C}_m\right)\left(\mathrm{z}(\mathrm{x}_i)\mathrm{z}(\mathrm{x}_i)^\top - \mathrm{C}_m\right)^\top\right]\right\|_2 \\
&= \frac{1}{n^2}\left\|\sum_{i=1}^n \mathbb{E}\left[\|\mathrm{z}(\mathrm{x}_i)\|^2\,\mathrm{z}(\mathrm{x}_i)^\top\mathrm{z}(\mathrm{x}_i) - \mathrm{z}(\mathrm{x}_i)^\top\mathrm{z}(\mathrm{x}_i)\mathrm{C}_m - \mathrm{C}_m\mathrm{z}(\mathrm{x}_i)^\top\mathrm{z}(\mathrm{x}_i) + \mathrm{C}_m^2\right]\right\|_2 \\
&\leq \frac{1}{n^2}\left\|\sum_{i=1}^n \mathbb{E}\left[\tau^2\mathrm{z}(\mathrm{x}_i)^\top\mathrm{z}(\mathrm{x}_i) - \mathrm{z}(\mathrm{x}_i)^\top\mathrm{z}(\mathrm{x}_i)\mathrm{C}_m - \mathrm{C}_m\mathrm{z}(\mathrm{x}_i)^\top\mathrm{z}(\mathrm{x}_i) + \mathrm{C}_m^2\right]\right\|_2 \\
&= \frac{1}{n^2}\left\|\sum_{i=1}^n \tau^2\mathrm{C}_m - \mathrm{C}_m^2 - \mathrm{C}_m^2 + \mathrm{C}_m^2\right\|_2 \\
&= \frac{1}{n}\left\|\tau^2\mathrm{C}_m - \mathrm{C}_m^2\right\|_2 \\
&\leq \frac{\tau^2}{n}\|\mathrm{C}_m\|_2 \\
&\leq \frac{\tau^2\lambda_1}{n}
\end{aligned}
$$

where the second last inequality holds because $\mathrm{C}_m$ is a positive semi-definite matrix. From matrix Bernstein concentration (Lemma F.2, restated from [Tropp et al., 2015]), we have, with probability at least $1 - \delta$

$$
\begin{aligned}
\left\|\widehat{\mathrm{C}}_m - \mathrm{C}_m\right\|_2 = \left\|\sum_1^m \Xi_i\right\|_2 &\leq \frac{L(\Xi)}{6}\log\left(\frac{\delta}{2m}\right) + \sqrt{\frac{L(\Xi)^2}{12}\log\left(\frac{\delta}{2m}\right)^2 + \log\left(\frac{\delta}{2m}\right)v(\Xi)} \\
&\leq \frac{\tau^2}{3n}\log\left(\frac{\delta}{2m}\right) + \sqrt{\frac{\tau^2}{3n}\log\left(\frac{\delta}{2m}\right)^2 + \log\left(\frac{\delta}{2m}\right)\frac{\tau^2\lambda_1}{n}}
\end{aligned}
$$

$\square$

In the following lemma, we show that $\mathcal{A}_{ERM}$ is an efficient subspace learner.

**Lemma D.2.** $\mathcal{A}_{ERM}$ *is an effcient subspace learner.*

*Proof.* We invoke Theorem F.1 with the sub-multiplicative norm being the spectral norm. With $\mathrm{A} = \mathrm{C}_m, \mathrm{B} = \widehat{\mathrm{C}}_m, \mathrm{U} = \widehat{\Phi}, \mathrm{V} = \Phi_k^\perp$. Let $\epsilon = \left\|\mathrm{C}_m - \widehat{\mathrm{C}}_m\right\|_2$. .From Weyl's inequality, we have $\lambda_k(\widehat{\mathrm{C}}_m) \geq \lambda_k - \epsilon = \lambda_{k+1} + \mathrm{gap} - \epsilon \geq \lambda_{k+1}$, if $\epsilon < \mathrm{gap}$. Therefore, setting $\mu = \lambda_{k+1}$, and

$\alpha = \text{gap} = \lambda_k - \lambda_{k+1}$, then with probability $1 - \delta$, we get

$$\left\| (\Phi_k^\perp)^\top \widehat{\Phi} \right\|_F^2 \le k \left\| (\Phi_k^\perp)^\top \widehat{\Phi} \right\|_2^2 \le \frac{k\epsilon}{\alpha^2}$$

$$\le \frac{k}{\alpha^2} \left( \frac{\tau}{3n} \log\left( \frac{\delta}{2m} \right) + \sqrt{\frac{\tau}{3n} \log\left( \frac{\delta}{2m} \right)^2 + \log\left( \frac{\delta}{2m} \right) \frac{\tau \lambda_1}{n}} \right)^2$$

$$\le \frac{k}{\alpha^2} \left( \frac{\tau}{3n} \log\left( \frac{\delta}{2m} \right) + \sqrt{\frac{2\lambda_1 \tau}{n} \log\left( \frac{\delta}{2m} \right)} \right)^2$$

$$\le \frac{k}{\alpha^2} \left( \frac{\lambda_1 \tau^2}{n} \log\left( \frac{\delta}{2m} \right)^2 \right)$$

Setting $q_{ERM}(1/\delta, \log(m), \log(n)) = \frac{\lambda_1 \tau^2}{\alpha^2} \log\left( \frac{\delta}{2m} \right)^2 = \frac{k \lambda_1 \tau^2}{(\lambda_k - \lambda_{k+1})^2} \log\left( \frac{\delta}{2m} \right)^2$, we get,

$$\left\| (\Phi_k^\perp)^\top \widehat{\Phi} \right\|_F^2 \le \frac{q_{ERM}(1/\delta, \log(m), \log(n))}{n}$$

$\square$

**Space and Computational Complexity of ERM:** ERM requires computing and storing the empirical covariance matrix $\widehat{C}_m$, which takes $O(m^2)$ memory. A rank $k$ SVD on $\widehat{C}_m$, generally, takes $O(m^2 k)$ computations. We note that there are methods to scale this up but it is out of the scope of this work.

### D.2 Oja's Algorithm

Having shown that ERM achieves optimal statistical rates, we now discuss a (relatively) more efficient algorithm in terms of space and computational complexity. We leverage the recent analysis of the classical Oja's algorithm and show how the algorithmic parameters affect the main result. We first restate the theorem statement from the analysis of Oja in Allen-Zhu and Li [2016a].

**Theorem D.3.** *Let* $\text{gap} := \lambda_k - \lambda_{k+1} \in \left( 0, \frac{1}{k} \right]$ *and* $\Lambda := \sum_{i=1}^{k} \lambda_i \in (0, 1]$, *for every* $\epsilon, \delta \in (0, 1)$ *define learning rates*

$$T_0 = \Theta\left( \frac{4k\Lambda}{\text{gap}^2 \delta^2} \right), T_1 = \Theta\left( \frac{\Lambda}{\text{gap}^2} \right), \eta_t = \begin{cases} \Theta\left( \frac{1}{\text{gap} \, T_o} \right) & 1 \le t \le T_0 \\ \Theta\left( \frac{1}{\text{gap}^2 \, T_1} \right) & T_0 < t \le T_0 + T_1 \\ \Theta\left( \frac{1}{\text{gap}(t - T_0)} \right) & t > T_0 + T_1 \end{cases}$$

*Let $Z$ be the column orthonormal matrix consisting of all eigenvectors of $C_m$ with values no more than $\lambda_{k+1}$. Then the output $Q_T$ of the algorithm satisfies with at least $1 - \frac{\delta}{2}$,*

$$\text{for every } T = T_0 + T_1 + \Theta\left( \frac{T_1}{\epsilon} \right), \text{ it satisfies } \left\| Z^\top Q_T \right\|_F^2 \le \epsilon$$

The above theorem gives guarantees of the form required by the definition of efficient subspace learner. Therefore, implicitly, Oja is an efficient subspace learner. This is formally stated in the following lemma.

**Lemma D.4.** $\mathcal{A}_{oja}$ *is an Efficient Subspace Learner.*

*Proof.* From Theorem D.3, we have

$$\left\|\mathbf{Z}^\top \mathbf{Q}_n\right\|_F^2 \leq \epsilon$$

$$= \tilde{\Theta}\left(\frac{T_1}{n - T_0 - T_1}\right)$$

$$\leq \tilde{\Theta}\left(\frac{2\Lambda}{\text{gap}^2\, n}\right) \qquad \text{(for large } n\text{)}$$

Setting $q_{oja}(1/\delta, \log(m), \log(n)) = \tilde{\Theta}\left(\frac{\Lambda}{\text{gap}^2}\right)$, we get,

$$\left\|\mathbf{Z}^\top \mathbf{Q}_n\right\|_F^2 \leq \frac{q_{oja}(1/\delta, \log(m), \log(n))}{n}$$

$\square$

Moreover the requirement of an initial constant number of samples as stated in Theorem 4.2 also appears in Theorem D.3 as warm-up phase. Therefore, the requirement of initial samples can be absorbed in the warm-up phase of Oja.

**Space and Computational Complexity of Oja's Algorithm:** Oja's algorithm takes $O(mk)$ memory. The per iteration computational cost is $O(mk)$. Therefore, for an $\epsilon$-suboptimal solution, the total computational cost is $O\left(\frac{mk}{\epsilon^2}\right)$.

# E  Experiments

We now need some lemmas which gives us analytical forms which would be used to calculate the objective with respect to empirical measure in the experiments.

Let $\widehat{P}_{\mathcal{A}}$ be the output of an efficient subspace learner $\mathcal{A}$. Let $\widehat{P}_{\mathcal{A}} = \tilde{\Phi}\tilde{\Phi}^{\top}$ be its eigendecomposition. We define $\widehat{\Phi} = \tilde{\Phi}R^*$, where let

$$R^* = \underset{R^\top R = RR^\top = I}{\arg\min} \left\| \tilde{\Phi}R - \Phi_k \right\|_{\mathcal{F}}^2$$

The following gives gives an explit form for $R^*$.

**Lemma E.1.** *For any orthogonal matrix $\tilde{\Phi} \in \mathbb{R}^{m \times k}$ and $\Phi \in \mathbb{R}^{m \times k}$, the solution of the optimization problem*

$$\underset{R^\top R = RR^\top = I}{\arg\min} \left\| \tilde{\Phi}R - \Phi \right\|_{\mathcal{F}}^2$$

*is* $R^* = \tilde{\Phi}^{\top}\Phi$

*Proof.*

$$\underset{R^\top R = RR^\top = I}{\arg\min} \left\| \tilde{\Phi}R - \Phi_k \right\|_{\mathcal{F}}^2 = \underset{R^\top R = RR^\top = I}{\arg\min} -\mathrm{Tr}\left( R^\top \tilde{\Phi}^\top \Phi_k \right)$$

$$\underset{R^\top R = RR^\top = I}{\max} \mathrm{Tr}\left( R^\top \tilde{\Phi}^\top \Phi_k \right) \leq \|R\|_F \left\| \tilde{\Phi}^\top \Phi_k \right\|_F = k$$

Note that $\mathrm{Tr}\left( R^* \tilde{\Phi}^\top \Phi_k \right) = k$. So, the maximum is achieved at $R = R^* = \tilde{\Phi}^\top \Phi_k$. $\qquad\square$

We have $\widehat{\Phi} = \tilde{\Phi}R^*$. We now use this and apply Lemma E.2 to evaluate the objective.

**Lemma E.2.** *For a projection matrix $P = UU^\top = \sum_{i=1}^{k} u_i \otimes_{\mathcal{F}} u_i$, $\langle \mathfrak{A}P, \mathfrak{I}C \rangle_\rho = \frac{1}{n}\mathrm{Tr}\left( V^\top KV \right)$, where $V = \Phi^\top US^{-\frac{1}{2}}$ and $S = \mathrm{diag}\left( \lambda_1, \lambda_2, \ldots, \lambda_k \right), \lambda_i = \langle C_m u_i, u_i \rangle_{\mathcal{F}}$*

*Proof of Lemma E.2.*

$$\langle \mathfrak{A}P, \mathfrak{I}C \rangle_{HS(\rho)} = \left\langle \sum_{i=1}^{k} \frac{Au_i}{\sqrt{\lambda_i}} \otimes_\rho \frac{Au_i}{\sqrt{\lambda_i}}, \sum_{j=1}^{n} \bar{\phi}_j \otimes_\rho \bar{\phi}_j \right\rangle_{HS(\rho)}$$

$$= \sum_{i,j=1}^{k,n} \frac{1}{\lambda_i} \left\langle Au_i, \bar{\phi}_j \right\rangle_\rho^2$$

$$= \sum_{i,j=1}^{k,n} \frac{1}{\lambda_i} \left\langle u_i, A^* \bar{\phi}_j \right\rangle_{\mathcal{F}}^2$$

where the second equality follows from bi-linearity of inner products, third from the definition of adjoints.

$$\left\langle u_i, A^* \bar{\phi}_j \right\rangle_{\mathcal{F}} = \sum_{l=1}^{m} (u_i)_l \left( A^* \bar{\phi}_j \right)_l$$

$$= \sum_{l=1}^{m} (u_i)_l \frac{1}{n} \sum_{q=1}^{n} \bar{\phi}_j(x_q)z(x_q)_l = \frac{1}{n} u_i^\top \Phi \bar{\Phi}_j$$

where $\bar{\Phi}_j \in \mathbb{R}^n$ and $\left( \bar{\Phi}_j \right)_q = \bar{\phi}_j(x_q)$.

Note that

$$\bar{\lambda}_j = \left\langle C\bar{\phi}_j, \bar{\phi}_j \right\rangle_{\mathcal{H}} = \left\langle I^* I\bar{\phi}_j, \bar{\phi}_j \right\rangle_{\mathcal{H}}$$
$$= \left\langle I\bar{\phi}_j, I\bar{\phi}_j \right\rangle_{\rho} = \left\langle \bar{\phi}_j, \bar{\phi}_j \right\rangle_{\rho}$$
$$= \frac{1}{n} \sum_{q=1}^{n} \bar{\phi}_j(x_q)^2 = \frac{1}{n} \left\| \bar{\Phi}_j \right\|_2^2$$

where the third equality follows from the property of adjoints.
Moreover, $\left(V_j^*\right)^{\top} K V_j^* = \bar{\lambda}_j$. Therefore $\bar{\Phi}_j^{\top} \bar{\Phi}_j = n\bar{\lambda}_j = n \left(V_j^*\right)^{\top} K V_j^*$.
So we have $\bar{\Phi}_j = \sqrt{n} K^{1/2} V_j^*$. Hence,

$$\langle \mathfrak{A}P, \mathfrak{J}C \rangle_{HS(\rho)} = \sum_{i,j=1}^{k,n} \frac{1}{\lambda_i} \left( \frac{1}{n} u_i^{\top} \Phi \sqrt{n} K^{1/2} V_j^* \right)^2$$
$$= \frac{1}{n} \sum_{i,j=1}^{k,n} \frac{1}{\lambda_i} \left( u_i^{\top} \Phi K^{1/2} V_j^* \right)^2$$
$$= \frac{1}{n} \sum_{i,j=1}^{k,n} \left( \frac{1}{\sqrt{\lambda_i}} u_i^{\top} \Phi K^{1/2} V_j^* \right)^2$$
$$= \frac{1}{n} \sum_{i,j=1}^{k,n} \left( V_i^{\top} K^{1/2} V_j^* \right)^2$$

where $V = \Phi^{\top} U S^{-\frac{1}{2}}$. Therefore, we have,

$$\langle \mathfrak{A}P, \mathfrak{J}C \rangle_{HS(\rho)} = \frac{1}{n} \sum_{i,j=1}^{k,n} \mathrm{Tr} \left( V_i^{\top} K^{1/2} V_j^* \right)^2$$
$$= \frac{1}{n} \sum_{i,j=1}^{k,n} \mathrm{Tr} \left( V_i^{\top} K^{1/2} V_j^* (V_j^*)^{\top} K^{1/2} V_i \right)$$
$$= \frac{1}{n} \sum_{i=1}^{k} \mathrm{Tr} \left( V_i^{\top} K^{1/2} \sum_{j=1}^{n} V_j^* (V_j^*)^{\top} K^{1/2} V_i \right)$$
$$= \frac{1}{n} \sum_{i=1}^{k} \mathrm{Tr} \left( V_i^{\top} K^{1/2} V^* (V^*)^{\top} K^{1/2} V_i \right)$$
$$= \frac{1}{n} \sum_{i=1}^{k} \mathrm{Tr} \left( V_i^{\top} K V_i \right)$$
$$= \frac{1}{n} \mathrm{Tr} \left( V^T K V \right)$$

$\square$

# F   Auxillary Results

Here we state some Auxillary results used in the proofs.

**Theorem F.1** (Generalized Gap free Wedin Theorem). *For $\epsilon > 0$, let A and B be two PSD matrices. For every $\mu > 0, \alpha > 0$, let U be column orthonormal matrix consisting of eigenvectors of A with eigenvalue $\leq \mu$, let V be column orthonormal matrix consisting of eigenvectors of B with eigenvalue $\geq \mu + \alpha$, then we have*

$$\left\| U^\top V \right\| \leq \frac{\|A - B\|}{\alpha}$$

*where the norm $\|\cdot\|$ is any sub-multiplicative norm.*

*Proof.* The above theorem is stated in [Allen-Zhu and Li, 2016b, Lemma B.3] in the sense of spectral norm. For the sake of completeness, we present the proof and show that it can easily be generalized to any sub-multiplicative norm.

Let the SVD of A and B be $A = U\Sigma U^\top + U'\Sigma'U'^\top, B = V\tilde{\Sigma}V^\top + V'\tilde{\Sigma}'V'^\top$, where $\Sigma$ is a diagonal matrix which contains all eigenvalues of A which are $\leq \mu$. Similarly, $\tilde{\Sigma}$ contains all eigenvalues $\geq \mu + \alpha$. Let $E := A - B$.

$$\Sigma U^\top = U^\top A = U^\top (B + E)$$

where the first equality follows because U is orthogonal to U'. Multiply by V on the right on both sides, we get,

$$\Sigma U^\top V = U^\top BV + U^\top EV = U^\top V\tilde{\Sigma} + U^\top EV$$

where the second equality follows because V is orthogonal to V'. Multiplying by $\tilde{\Sigma}^{-1}$ on the right on both sides, we get,

$$\Sigma U^\top V \tilde{\Sigma}^{-1} = U^\top V + U^\top EV\tilde{\Sigma}^{-1}$$

Taking any sub-multiplicative norm on the left hand side, we obtain an upper bound on it as follows,

$$\left\| \Sigma U^\top V \tilde{\Sigma}^{-1} \right\| \leq \|\Sigma\|_2 \left\| \tilde{\Sigma}^{-1} \right\|_2 \left\| U^\top V \right\|$$
$$\leq \frac{\mu}{\mu + \alpha} \left\| U^\top V \right\|$$

where the first inequality follows from the property of sub-multiplicative norms, and the second from the definition of $\Sigma$ and $\tilde{\Sigma}$.

Similarly, taking any sub-multiplicative norm on the right hand side, we get a lower bound on it as follows,

$$\left\| U^\top V + U^\top EV\tilde{\Sigma}^{-1} \right\| \geq \left\| U^\top V \right\| - \left\| U^\top EV\tilde{\Sigma}^{-1} \right\|$$
$$\geq \left\| U^\top V \right\| - \left\| U^\top \right\|_2 \|E\| \|V\|_2 \left\| \tilde{\Sigma}^{-1} \right\|_2$$
$$\geq \left\| U^\top V \right\| - \frac{\|E\|}{\mu + \alpha}$$

where the first inequality follows from (reverse) triangle inequality, the second from property of sub-multiplicative norms and third because U and V are orthonormal matrices and by definition of $\tilde{\Sigma}$. Combining both the bounds, we get,

$$\left\| U^\top V \right\| \left( 1 - \frac{\mu}{\mu + \alpha} \right) \leq \frac{\|E\|}{\mu + \alpha}$$
$$\implies \left\| U^\top V \right\| \leq \frac{\|E\|}{\alpha}$$

$\square$

**Theorem F.2** (Matrix Bernstein [Tropp et al., 2015])**.** *Let* $S_1, S_2, \ldots S_n$ *be* $n$ *i.i.d* $d_1 \times d_2$ *random matrices such that* $\mathbb{E}S_i = 0, \|S_i\| \leq L \; \forall i \in [n]$. *Let* $Z = \sum_{i=1}^{n} S_i$. *Let* $v(Z)$ *denote the matrix variance statistic of the sum defined as,*

$$v(Z) = \max\{\mathbb{E}ZZ^\top, \mathbb{E}Z^\top Z\}$$

*Then, with probability at least* $1 - \delta$*, we have,*

$$\mathbb{P}\{\|Z\| \geq t\} \leq (d_1 + d_2) \exp\left(\frac{-t^2/2}{v(Z) + Lt/3}\right) \; \forall t \geq 0$$

**Theorem F.3** (Local Rademacher Complexity [Bartlett et al., 2002])**.** *Let* $\mathcal{X}$ *be a measurable space. Let* $\mathcal{P}$ *be a probability distribution on* $\mathcal{X}$ *and let* $x_1, x_2 \ldots x_n$ *be i.i.d. samples drawn from* $\mathcal{P}$*. Let* $\mathcal{P}_n$ *denote the empirical measure. Let* $\mathcal{F}$ *be a class of functions on* $\mathcal{X}$ *ranging from* $[-1, 1]$ *and assume that there exists some constant* $B$ *such that for every* $f \in \mathcal{F}, \mathcal{P}^2 f \leq B\mathcal{P}f$. *Let* $\psi$ *be a sub-root function and let* $r^*$ *be the fixed point of* $\psi$*. If* $\psi$ *satisfies*

$$\psi(r) \geq B\mathbb{E}_{X,\sigma}\left[\mathcal{R}_n\{f \in star(\mathcal{F})|\mathcal{P}f^2 \leq r\}\right]$$

*where* $star(\mathcal{F}) = \{\lambda f | f \in \mathcal{F}, \lambda \in [0,1]\}$ *is the star shaped hull of* $\mathcal{F}$ *and* $\mathcal{R}_n\mathcal{F} = \sup_f \in \mathcal{F}\frac{1}{n}\sum_{i=1}^{n}\sigma_i f(x_i)$ *is the empirical Rademacher complexity of* $\mathcal{F}$ *given data points* $\{x_i\}_{i=1}^{n}$*; then for every* $K > 0$ *and* $x > 0$*, with probability at least* $1 - e^{-\delta}$

$$\forall f \in \mathcal{F}, \mathcal{P}f \leq \frac{K}{K-1}\mathcal{P}_n f + \frac{6K}{B}r^* + \frac{\delta(11 + 5BK)}{n} \tag{6}$$

*Also, with probability at least* $1 - e^{-\delta}$

$$\forall f \in \mathcal{F}, \mathcal{P}_n f \leq \frac{K}{K+1}\mathcal{P}f + \frac{6K}{B}r^* + \frac{\delta(11 + 5BK)}{n} \tag{7}$$

*Furthermore, if* $\widehat{\psi}_n$ *is a data-dependent sub-root function with fixed point* $\widehat{r}^*$ *such that*

$$\psi^*(r) > 2(10 \vee B)\mathbb{E}_\sigma\left[\mathcal{R}_n\{f \in star(\mathcal{F})|\mathcal{P}^n f^2 \leq 2r\}\right] + \frac{2(10 \vee B + 11)\delta}{n}$$

*then with probability at least* $1 - 2e^\delta$*, it holds that* $\widehat{r}^* \geq r^*$*; as a consequence, equations 6 and 7 holds with* $r^*$ *replaced by* $\widehat{r}^*$