[Reviews · NeurIPS 2018]

Reviewer 1



The paper considers the problem maximizing the expectation of ||Qx||^2 over n-by-k matrices Q whose columns are mutually orthogonal unit vectors, where x is a random vector from a given distribution. The main results are: (i) sub-quadratic time algorithm for computing the ERM (ii) generalization for streaming data (iii) theoretical new connections between covariance operators in RKHS and the space of square integrable functions with respect to data distribution. Strong: - Interesting and non-trivial results and analysis. - Good experimental results. Weak: - The algorithm is very simple: just sample i.i.d. from "RandomFeatures". Do we really need all these theorems to obtain this result? - My main concern is that this result can be trivially obtained (including streaming) using coresets. Coresets are not mentioned at all in this paper and I suspect that the authors are not aware. See e.g. recent coreset and sketches papers for PCA by Christian Sohler, Michael Cohen, Mahoney, and Woodruf. They are for linear functions but it seems that generalization may not be hard. Please compare and cite.

Reviewer 2



Streaming Kernel PCA with Õ( n) Random Features comments: - there is a very related paper NIPS 3489 - the theory seems to be ok but is also widely following concepts provided already by the cited papers (mainly along work of Oja, Mahoney,Blanchard and others) - I am very unhappy with the experimental part - very small - From the paper it is unclear how the Nystroem method works in this context and how this fits in. The Nystroem method is defined for static (non-streaming) data distributions and used to approximate a psd similarity matrix I do not see how this fits in your scenario - nor how you could have applied the approach in a realistic comparison (no ! details are given) Comments to author feedback: - thanks for clarifying some issues - I am still a bit unhappy with some underlying assumptions and the link to Nystroem - I think the paper would be much better if the theoretical findings are supported by some results which would also show that numerical issues do not negatively effect the theoretical claims.

Reviewer 3



=== SUMMARY === The paper deals with efficient computation of kernel PCA using random Fourier features. It proves that under a certain distributional assumption (of exponential spectrum decay) on the input data, ~O(sqrt(n)) random features suffice for a KPCA algorithm to converge to optimum. The framework is instantiated with two algorithms: ERM and Oja's algorithm. === COMMENTS === The topic of the paper is significant and lies in the mainstream of machine learning. The improvement in the asymptotic bound on the number of features is substantial and the theoretical content looks solid. The mathematical results are presented in detail, with all their assumptions and subtleties made explicit and verifiable. The authors highlight that the proposed method does not yield a projection of the kernel space, but an operator which is asymptotically close to a projection in a formal sense. However it is not clear how to interpret it, and some discussion could help. What guarantees do we have on the output operator? Do other methods (except ERM) have a similar limitation? How does this discrepancy affect the downstream uses of KPCA? This question arises in light of the experimental results, where the proposed methods seem to produce solutions which are apparently far from being feasible (for eq.(2)) whenever they don't coincide with the baselines (more on this later). The mathematical part (which forms the bulk of the paper) somewhat lacks in clarity. There is a lot of heavy notation introduced, and some of it seems overly cumbersome and possibly too general for the needs of the paper. An effort to simplify it to the extent possible would help improve readability. The text does not survey the proof techniques or distinguish a novel technical insight from the mundane derivations. This makes it difficult to gauge the level of insight and sophistication of the ideas used in the proofs. A roadmap of the proof or an explicit survey of the claimed "crucial and novel insights" (line 76) would help here, especially since they are claimed to be of potentially more general use; it is currently not evident what they are. On a related note, a comparison with the techniques of prior works on the subject (eg. [RR17] which is cited as a precursor) might be in order, if relevant. Some more minor clarity issues: 1. The notion of streaming KPCA is stated in the title but not properly explained or referenced in the text. 2. Table 1 is written in terms of epsilon while the text is parameterized by n,m. This epsilon is not defined or explained, even though it is mostly understood by context. Also the table does not address the limitation of the listed methods (apparently, as per above, some of them do not formally solve the KPCA problem as formulated in eq.(2)). I found the experimental section quite puzzling on several counts. I might be misunderstanding it, and I hope the authors somewhat clarify this in their response. -- You write that ERM is slower by an order of magnitude compared to the other three methods; where do you see this? The running time plots look like all four methods converge within almost exactly the same time, and that ERM is even slightly better than Nystrom. The proposed methods RF-ERM and RF-Oja begin to produce data points much earlier, but it is not at all clear what this means: their objective value is much higher than the maximum, which apparently implies that they are pretty far from being projections, and as written above I am not sure how to interpret this relaxed notion of solution and whether it is usable as output. -- What are we supposed to see in the iteration plots? In what sense are the curves comparable? Do they measure the number of data point samples until convergence? And don't they show (in whatever metric they measure) that RF-Oja is significantly inferior to the two baselines (while RF-ERM is just slightly better)? -- Why are the proposed methods compared only to baselines and not to competitive prior work (such as RF-DSG from Table 1)? -- In general, is the purpose of this section to make a case for the usability of the proposed methods, or just to verify the theoretical predictions of Corollary 4.4? === CONCLUSION === Pros: 1. The paper deals with an important and fundamental problem, and seems to make solid progress on its theoretical understanding. 2. The mathematical analysis is detailed, elaborate and well-formalized. Cons: 1. The paper could be improved in clarity of presentation of the mathematical content. The current form might inhibit its accessibility to wider NIPS audience. The results are clearly stated but the techniques are not as well explained. 2. The relaxed notion of solution used by the proposed methods (which is an apparent limitation) could be better discussed in terms of its meaning and downstream applicability. 3. The experimental section is not entirely clear, and in particular does not make a solid case for the empirical usability of the proposed method. === UPDATE AFTER AUTHOR RESPONSE === Thank you for the clarifications. I would encourage to include some of them in the paper, especially w.r.t the experimental section which became much clearer to me. All suggestions on presentation are left to the authors' discretion; the one I would strongly promote is mentioning the assumptions/limitations of the listed methods in Table 1, so as to maintain formal correctness in the comparison to prior work.